# Mitochondrially-targeted APOBEC1 is a potent mtDNA mutator affecting mitochondrial function and organismal fitness in *Drosophila*

Simonetta Andreazza [1], Colby L. Samstag [2], Alvaro Sanchez-Martinez [1], Erika Fernandez-Vizarra [1], Aurora Gomez-Duran [1], Juliette J. Lee[1], Roberta Tufi [1], Michael J. Hipp[3], Elizabeth K. Schmidt[3], Thomas J. Nicholls[1], Payam A. Gammage [1], Patrick F. Chinnery[1,4], Michal Minczuk [1], Leo J. Pallanck[2], Scott R. Kennedy [3] & Alexander J. Whitworth [1]

Somatic mutations in the mitochondrial genome (mtDNA) have been linked to multiple disease conditions and to ageing itself. In *Drosophila*, knock-in of a proofreading deficient mtDNA polymerase (*POLG*) generates high levels of somatic point mutations and also small indels, but surprisingly limited impact on organismal longevity or fitness. Here we describe a new mtDNA mutator model based on a mitochondrially-targeted cytidine deaminase, APOBEC1. *mito-APOBEC1* acts as a potent mutagen which exclusively induces C:G>T:A transitions with no indels or mtDNA depletion. In these flies, the presence of multiple non-synonymous substitutions, even at modest heteroplasmy, disrupts mitochondrial function and dramatically impacts organismal fitness. A detailed analysis of the mutation profile in the *POLG* and *mito-APOBEC1* models reveals that mutation type (quality) rather than quantity is a critical factor in impacting organismal fitness. The specificity for transition mutations and the severe phenotypes make *mito-APOBEC1* an excellent mtDNA mutator model for ageing research.

[1] MRC Mitochondrial Biology Unit, University of Cambridge, Cambridge Biomedical Campus, Hills Road, Cambridge CB2 0XY, UK. [2] Department of Genome Sciences, University of Washington, Seattle, WA 98195, USA. [3] Department of Pathology, University of Washington, Seattle, WA 98195, USA. [4] Department of Clinical Neuroscience, School of Clinical Medicine, University of Cambridge, Cambridge CB2 0QQ, UK. Correspondence and requests for materials should be addressed to A.J.W. (email: a.whitworth@mrc-mbu.cam.ac.uk)

Mitochondria play essential roles in many cellular metabolic and signalling processes, most notably in the production of molecular energy in the form of ATP and in the regulation of cell death. Mitochondria also contain their own genome (mtDNA)—in animals, a circular DNA molecule that contains 37 genes encoding 13 subunits of the respiratory chain and ATP synthase, along with the tRNAs and rRNAs necessary for their production[1]. As such, the transmission of germline mtDNA mutations, which occurs exclusively via the maternal lineage, can give rise to devastating mitochondrial diseases[2,3]. Furthermore, since mtDNA exists in multiple copies per mitochondrion, and thus many copies per cell, this can give rise to a mixed population of wild-type and mutated genomes, a condition known as heteroplasmy.

MtDNA has long been considered to be particularly vulnerable to accumulating spontaneous mutations due to its high replication rate, proximity to the major source of reactive oxygen species (the respiratory chain), the absence of protective histones and limited repair mechanisms[4]. Consistent with this idea, mtDNA mutations have been seen to accumulate with age in somatic tissues in a wide range of organisms including humans, rodents and invertebrates, and thus have been implicated as a driving force in the ageing process[5,6]. Moreover, high levels of mtDNA mutations have also been found in various age-related conditions including neurodegenerative disorders such as Parkinson's and Alzheimer's diseases[7,8].

To explore the consequences of increasing the level of mtDNA mutations on cellular and organismal fitness, and the mechanisms that counteract such mutations, a number of model systems have been developed. The best-established model system for inducing high levels of mtDNA point mutations is via the expression of a proofreading (exonuclease) deficient variant of the sole DNA polymerase responsible for replicating mtDNA, POLG. Mouse strains of this model system exhibit many characteristics of a premature ageing syndrome, including kyphosis, alopecia, and osteoporosis, all correlating with a shortened lifespan[9,10]. However, the mutational heterogeneity that arises in this model has led to considerable debate about the pathogenic entity and whether point mutations are actually driving ageing[11–14].

Equivalent 'mutator' models have been established in Drosophila, either by homologous recombination targeting the endogenous locus of the POLG homologue, tamas (tamas[exo−] flies)[15], or using a genomic transgene bearing a defective tamas[16]. Recent studies have shown that mutations that arise in these models have very limited impact on adult fly health or lifespan[16,17], questioning whether mtDNA mutations are capable of rising to levels sufficient to impact ageing in flies[17].

To circumvent the limitations of the POLG model, we sought to develop an alternative mtDNA mutator system by employing the activity of the cytidine deaminase APOBEC1. APOBEC1 (Apolipoprotein B (apoB) mRNA editing catalytic polypeptide 1) is a vertebrate-specific, zinc-dependent deaminase which, in complex with complementing specificity factor (ACF), catalyses the deamination of cytosine to uracil (C>U) in apoB mRNA[18–20]. Despite a physiological role in mRNA editing, which is fully dependent upon an RNA-binding accessory subunit, APOBEC1 alone was found to be a potent mutator of DNA both in vitro[21] and in vivo[22]. APOBEC1's effects as a DNA mutator are becoming more widely recognised and this ability is being exploited as a targeting mutagen in combination with zinc-finger peptides and CRISPR/Cas9[23,24]. Mechanistically, uracil (U) present in DNA is recognised as mutagenic and is usually removed by uracil-DNA glycosylases, generating an abasic site, which can itself be inappropriately filled. However, if uncorrected, in subsequent replication adenine (A) is paired with U to cause a mutagenic transition of a C:G to U(T):A, resulting in a stable C:

G>T:A point mutation. Importantly, this phenomenon recapitulates the predominant mutation profile in human ageing[25]. Thus, we sought to exploit this deaminase activity, directing APOBEC1 specifically to mitochondria via a construct we termed mito-APOBEC1.

In contrast to heterozygous tamas[exo−] flies, somatic mito-APOBEC1 expression severely limits mitochondrial function, organismal vitality and lifespan. A detailed analysis of the mutation spectra in these models reveals that the mutation profile (i.e., quality), rather than the overall mutation load (quantity), correlates with a reduced organismal fitness.

## Results

**Analysis of tam[exo−] mtDNA mutagenesis.** A Drosophila model equivalent to the 'mutator' mouse has been generated by knock-in of a proofreading (exonuclease) deficient variant of the POLG homologue, tamas (tam[exo−])[15]. Surprisingly, tam[exo−] adults have shown very limited defects on organismal health or lifespan[17], prompting us to examine in greater detail their mutation spectra. Since the tam[exo−] strain is homozygous lethal, all our analyses were conducted on heterozygous animals, inherited either paternally (+/tam[exo−]) or maternally (tam[exo−]/+). Reinsertion of the wild-type coding region, referred to as tam[rescue], served as a control genotype. As an additional control we also compared the tam lines to a strain bearing the closest nuclear and mitochondrial genetic background; white[Dahomey] (w[Dah]) for maternal inheritance or white[1118] (w[1118]) for paternal inheritance (see Methods).

Sequence analysis of the tam[exo−] strains and controls has previously been performed using a PCR-based cloning and sequencing strategy[15,17]. We sought to extend this analysis by performing Duplex Sequencing of the mitochondrial genome on individual brains from mutator and control animals (see 'Methods' section). Duplex Sequencing is a high-accuracy next-generation sequencing approach capable of detecting a single mutation in >10[7] wild-type bases[26]. We initially examined the mtDNA mutation profile of 10-day-old flies to allow for a modest amount of post-mitotic accumulation of mtDNA mutations. Our sequencing analysis provided a depth of ~500–1000 unique reads per site. Although coverage is not uniform across the genome, patterns of sequencing depth were consistent across all samples tested, suggesting that no samples bore large deletions within mtDNA (Supplementary Fig. 1). Sequence analysis identified multiple mutations at varying levels of heteroplasmy and several homoplasmic polymorphisms, depending on the background strain (Supplementary Data 1). Because we believe high heteroplasmy mutations could contribute to organismal phenotypes, we considered a threshold of 70% heteroplasmy (the maximum we observed) so as to exclude homoplasmic sequence polymorphisms present in the background parental strain but capture all other mutations. Hence, for all our analyses we have included the total number of observed mutations, including multiple counts for mutations at a same site (heteroplasmy). Mutation loads were calculated by dividing all mutations by the total number of nucleotides sequenced, considering the whole or a subset of the genome, as relevant to each analysis.

The background mtDNA mutation load varied a little by genotype and individual animal (see Supplementary Data 1), being $1.5–17 \times 10^{-6}$ depending on background strain, but comparable with previous measurements[27]. The tam[rescue] controls exhibited similar levels of mtDNA mutations to the background controls (Supplementary Data 1). In animals with paternally inherited tam[exo−], and hence accumulating only somatic (and not germline-inherited) mutations, the mtDNA mutation load increased to $5.7 \times 10^{-5}$ (Fig. 1a). However, in

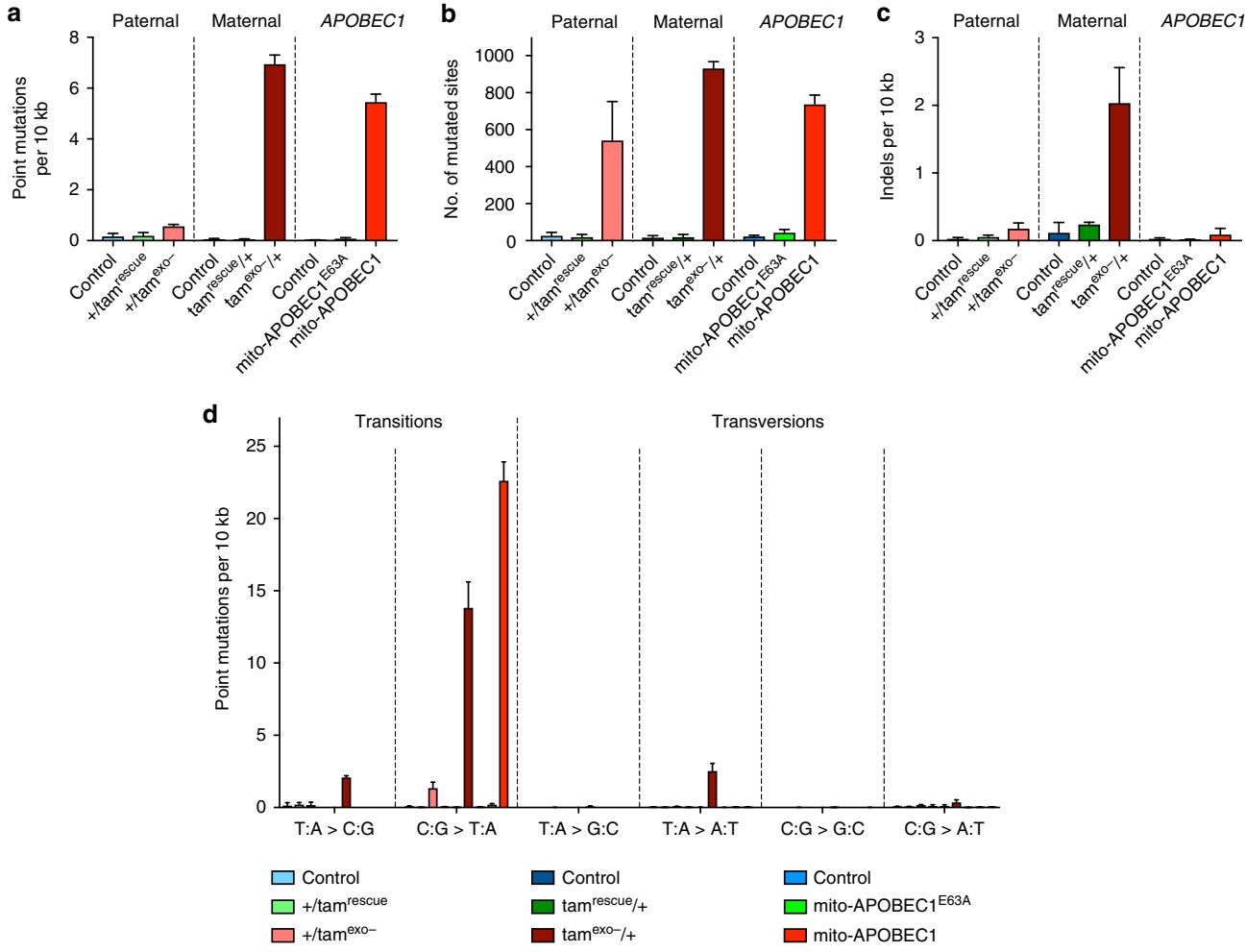

**Fig. 1** Maternally inherited *tam*exo− and *mito-APOBEC1* flies generate high mtDNA mutation level. Quantification of mtDNA mutations in 10-day-old paternally inherited *tam*exo− (+/*tam*exo−), maternally inherited *tam*exo− (*tam*exo−/+) and *mito-APOBEC1* mutator flies and respective controls. **a** mtDNA mutation load, expressed as number of mutations observed, including heteroplasmic mutations, per total number of bases sequenced [Note that coverage at the non-coding A+T region, because of its repetitive high A:T content, is minimal and therefore, negligibly contributes to the overall mutation load], **b** the number of sites with at least one mutation, **c** indel mutation load, and **d** the mutation load identified by mutation type as indicated. Charts show mean ± SD, n = 3 animals per genotype. Full genotypes are given in Methods. Source data are provided as a Source Data file and Supplementary Data 1 and 2

animals with maternally inherited *tam*exo−, mtDNA mutation levels were substantially higher ($6.9 \times 10^{-4}$; Fig. 1a). Importantly, maternally inherited *tam*exo− flies showed no significant alterations in mtDNA copy number (see below). The strong increase in mutation load of maternally inherited *tam*exo− is likely a combination of inherited heteroplasmic mtDNA copies from the *tam*exo−-bearing mothers and clonal expansion of replication errors occurring during the peak mtDNA replication that happens part-way through embryogenesis[28]. This is at least partially supported by the relatively similar number of sites (<2-fold difference) that are mutated in the paternally and maternally transmitted mutator flies (Fig. 1b). A much larger mutation load in a similar number of sites indicates that the same mutated sites are represented many more times in the maternal *tam*exo−/+. These results indicate that zygotically expressed (paternally inherited) heterozygous *tam*exo− is not a potent mutator, but mtDNA mutations can rise to high loads when *tam*exo− is maternally inherited. However, maternally inherited *tam*exo− also introduces a significant number of small insertion/deletion (indel) mutations (Fig. 1c, and Supplementary Data 2), which can confound the physiological interpretation of mtDNA point mutations. Thus, we sought to develop an alternative genetically

encoded mtDNA mutator system with a greater specificity for inducing point mutations.

**Generation of a new mtDNA mutator model**. We generated an inducible mutator model by exploiting the nucleic acid editing capability of the cytidine deaminase APOBEC1, which has been shown to cause C:G>T:A transition mutations in DNA[22]. Rat APOBEC1 was cloned into a 'UAS' transgenic expression vector with an N-terminal mitochondrial targeting sequence (MTS) followed by a hemagglutinin (HA) tag (Fig. 2a). A nuclear export sequence (NES) was also included to minimise any potential for nuclear localisation. We termed this construct '*mito-APOBEC1*'. As a control, we also generated an equivalent construct expressing a catalytically inactive mutant, E63A ('*mito-APOBEC1*E63A')[29]. Independent transgenic lines were established with the constructs in the same genomic locus to ensure comparable expression levels and genetic background (Fig. 2b). Subcellular fractionation analyses determined that *mito-APOBEC1* and *mito-APOBEC1*E63A were successfully targeted to mitochondria (Fig. 2c); this was further confirmed by immunofluorescence analysis in larval epidermal cells (Fig. 2d).

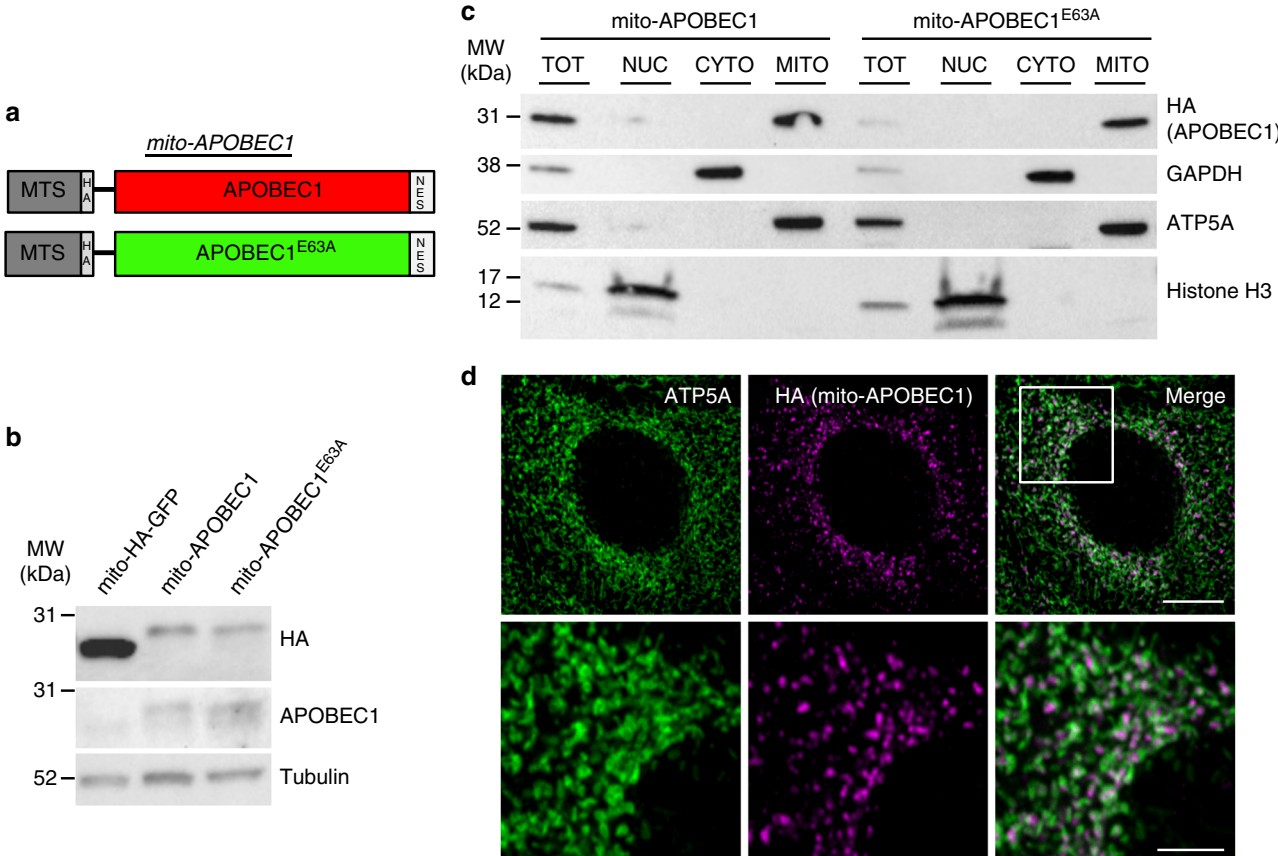

**Fig. 2** Generation and validation of the *mito-APOBEC1* mutator model. **a** Schematic of *mito-APOBEC1* construct and inactive mutant (E63A). **b** Immunoblot showing *mito-APOBEC1* transgene expression in whole fly extracts, using anti-APOBEC1 and anti-HA antisera. Transgenic *mito-HA-GFP* expression is used as a positive control, and anti-Tubulin immunostaining as loading control. **c** Tissue fractionation and immunoblotting shows that mito-APOBEC1 and mito-APOBEC1[E63A] proteins localise to the mitochondrial fraction (ATP5A-positive). GAPDH and Histone H3 mark the cytoplasmic and nuclear fractions, respectively. **d** Immunohistochemistry analysis of *mito-APOBEC1* expression in larval epidermal cells, revealed by anti-HA staining, co-stained with the mitochondrial membrane marker ATP5A. Scale bars = 10 μm (top panels) and 4 μm (bottom panels). Source data are provided as a Source Data file

Initial tests determined that the expression of *mito-APOBEC1* induced by a range of GAL4 drivers had no gross effects on development or viability. We chose to conduct the current study using the ubiquitous driver, *da-GAL4*. In all instances, transgene expression was only induced in the zygote, thus avoiding maternal transmission of germline mutations. Sequence analysis of mtDNA mutations revealed that zygotic expression of *mito-APOBEC1* is a highly effective mutagen, creating mutation loads of ~$5.5 \times 10^{-4}$, similar to maternally inherited *tam*[exo−], and substantially more effective than zygotically expressed, paternally inherited *tam*[exo−] (Fig. 1a and Supplementary Data 1). Interestingly, *mito-APOBEC1* mutates a similar number of mitochondrial genome sites as *tam*[exo−] (Fig. 1b). Also, *mito-APOBEC1* expression has minimal impact on mtDNA levels (Fig. 3b,d) and does not affect mtDNA integrity (Fig. 3c, e).

Analysing the mutation profile we found that *mito-APOBEC1*-induced changes were almost entirely composed of C:G>T:A transitions (Fig. 1d, and Supplementary Fig. 2a), consistent with its mechanism of action. In contrast, although the majority of *tam*[exo−] mutations were also C:G>T:A, a considerable proportion were T:A>C:G transitions or T:A>A:T transversions (Fig. 1d, and Supplementary Fig. 2a), as observed in the mouse[30]. Interestingly, mapping reciprocal mutations by strand showed a dramatic bias for *mito-APOBEC1*, with most of the cytidine deamination events occurring in the minor strand (G>A transitions in the major strand) (Supplementary Fig. 3a). This is in line with the strand pattern and type of mutations naturally occurring in *Drosophila*

during ageing and across generations[27,31]. Importantly, in contrast to maternally inherited *tam*[exo−], *mito-APOBEC1* does not induce small indels (Fig. 1c, and Supplementary Data 2), nor larger deletions (Fig. 3e). Taken together, these results show that *mito-APOBEC1* is a highly effective, inducible mtDNA mutator that exclusively causes C:G>T:A transitions at high loads without causing indel mutations or substantial mtDNA depletion.

**The two mutator models affect organismal fitness differently.** We then evaluated the impact of the mutator systems on organismal fitness. Foremost, with the long-standing association of mtDNA mutations and ageing, we assessed the effects on lifespan. In agreement with recent findings[17], we found that both paternally and maternally inherited *tam*[exo−] showed minimal impact on lifespan (Fig. 4a, c). In stark contrast, expression of *mito-APOBEC1* caused a dramatic reduction in median and maximal lifespan (Fig. 4e).

This striking difference in lifespan led us to investigate whether there was a similar differential effect on vitality during ageing as an indicator of 'healthspan'. Locomotor assays, such as startle-induced negative geotaxis (climbing), offer a sensitive and reliable read-out of the neuromuscular circuit function. Analysing the climbing ability of the *tamas* mutator lines, either paternally or maternally inherited, we found no difference between any of the genotypes in 2-, 10- and 30-day-old flies (Fig. 4b, d). However, consistent with the effects on lifespan, expression of

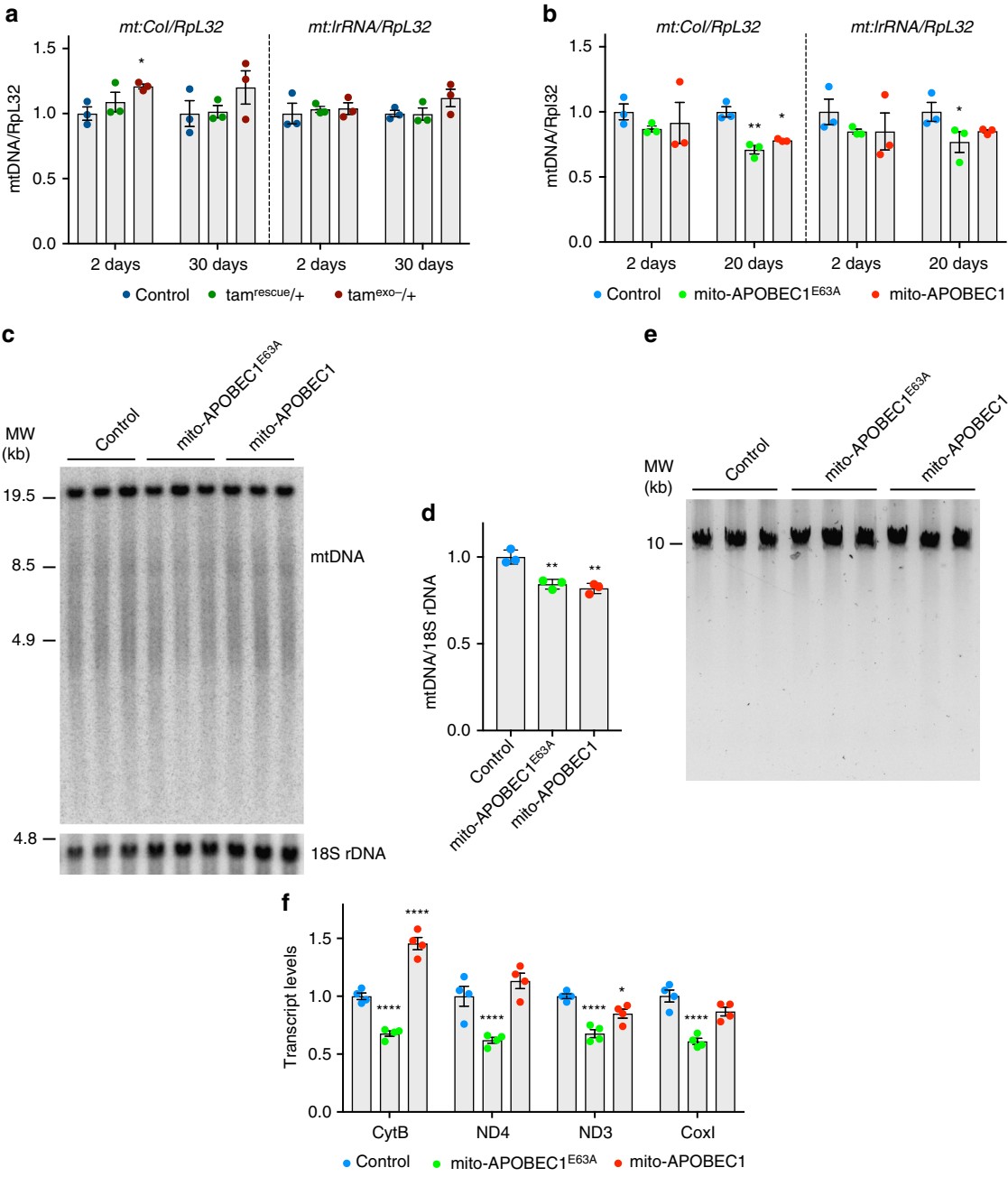

**Fig. 3** Analysis of mtDNA levels and integrity. mtDNA copy number was analysed in 10-day-old **a** maternally inherited *tam*[exo−] and **b** *mito-APOBEC1* mutator flies and respective controls by quantitative-PCR of two regions of the mitochondrial genome (*mt:CoI* and *mt:lrRNA*) against the level of nuclear gene, *RpL32*. Charts show means ± SEM, *n* = 3 biologically independent samples; data points indicate independent experiments. Statistical analysis used two-way ANOVA with Tukey's multiple comparison test. *$P < 0.05$, **$P < 0.01$. **c** Southern blot, quantified in (**d**), and **e** long-range PCR analyses further confirm mtDNA integrity and levels in *mito-APOBEC1* flies and relative controls. Charts in **d** show means ± SD, *n* = 3 biologically independent samples. Statistical analysis used one-way ANOVA with Sidak's multiple comparison test. *$P < 0.05$. **f** Levels of mitochondrial transcripts in one-week-old *mito-APOBEC1* flies and controls. Charts show means ± SEM, *n* = 4 biologically independent samples. Statistical analysis used two-way ANOVA with Tukey's multiple comparison test. *$P < 0.05$, ***$P < 0.001$, ****$P < 0.0001$. All other comparisons are non-significant but are not indicated to aid clarity. Source data are provided as a Source Data file

*mito-APOBEC1* caused a significant reduction in climbing ability across all ages tested (Fig. 4f).

Given the strong phenotypic effects of *mito-APOBEC1* but not *tam*[exo−], we sought to address potentially unrecognised extra-mitochondrial effects of this construct. To this end, we generated transgenic lines expressing HA-tagged wild-type and mutant *APOBEC1* variants lacking the MTS but retaining the NES

sequences (Supplementary Fig. 4a), which we termed 'cyto-APOBEC1' and 'cyto-APOBEC1[E63A]'. The *cyto-APOBEC1* lines appear to express the transgene at higher level than *mito-APOBEC1* lines, in comparison to *mito-HA-GFP* (compare Fig. 2b with Supplementary Fig. 4b); yet encouragingly, we observed no effect of *cyto-APOBEC1* expression upon lifespan or locomotor ability with age (Supplementary Fig. 4c, d). Thus, there appears to

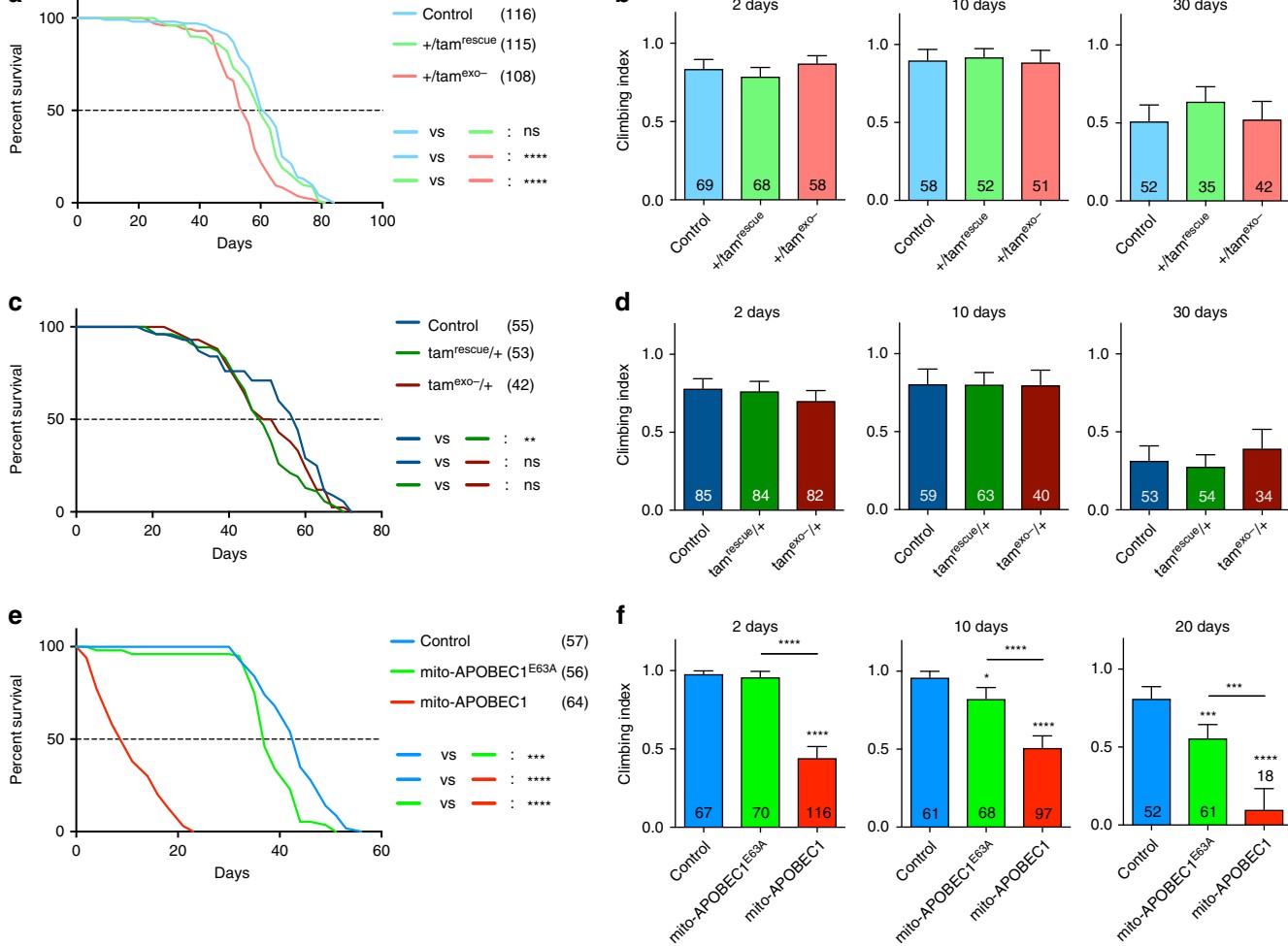

**Fig. 4** *mito-APOBEC1* but not *tam*$^{exo-}$ impacts fly lifespan and locomotor ability. **a**, **c**, **e** Lifespan and **b**, **d**, **f** locomotor (climbing) ability were assessed for **a**, **b** paternally inherited *tam*$^{exo-}$, **c**, **d** maternally inherited *tam*$^{exo-}$ and **e**, **f** *mito-APOBEC1* flies and respective controls. Only male flies were tested. For lifespan, statistical significance was determined by Log-rank (Mantel–Cox) test; the number of animals tested are indicated in the legend. For locomotor assays, charts show mean ± 95% confidence interval. The number of animals tested are shown per column. Statistical significance was determined by Kruskal–Wallis nonparametric test with Dunn's multiple comparisons correction. *$P < 0.05$, **$P < 0.01$, ***$P < 0.001$, ****$P < 0.0001$. All other comparisons are non-significant but are not indicated to aid clarity. Source data are provided as a Source Data file

be no detrimental effects of *cyto-APOBEC1*, substantiating the idea that the impact of *mito-APOBEC1* expression on organismal fitness derives exclusively from mtDNA mutations.

**mito-APOBEC1 but not tam$^{exo-}$ affects mitochondrial function.** Having established the relative impact of the mutator models on organismal fitness, we next sought to understand the impact on mitochondrial function. We first assessed the oxygen consumption rate (OCR) using high-resolution respirometry. Analysing the maternally inherited *tam*$^{exo-}$ heterozygotes we found no significant change to either Complex I- or Complex II-linked respiration, even after substantial ageing (Fig. 5a). In contrast, *mito-APOBEC1* expressing flies showed a significant decrease in Complex I-linked respiration, already in very young flies (Fig. 5b). Although the downturn in Complex II-linked respiration did not reach significance at 2 days, by 10 days old this too was significantly reduced (Fig. 5b). Because Complex-II is exclusively composed by nuclear-encoded proteins, we hypothesized the defect was due to dysfunctional Complex-III and/or IV. Consistent with these findings, BN-PAGE and in-gel activity assays showed a decrease in Complex-I and Complex-IV activity, while Complex-II activity was undiminished, even possibly increased

(Fig. 5c). Assessing the steady-state levels of the assembled complexes by native-PAGE immunoblotting we found that *mito-APOBEC1* flies consistently displayed decreased amounts of higher-order assemblies of Complexes I, III, IV and V (Fig. 5d), with partial accumulation of non-functional intermediates.

MtDNA mutations could potentially also impact transcription, causing a decline in transcript levels. To assess this, we analysed several transcripts from the mitochondrial genome by quantitative RT-PCR. Expression of the catalytically dead *mito-APOBEC1*$^{E63A}$ resulted in a moderate but significant reduction of all mitochondrial transcripts analysed (Fig. 3f), which mirrored the lowered amounts of mtDNA copies in these flies (Fig. 3b, d). In contrast, transcript levels in *mito-APOBEC1* flies were comparable or slightly higher than controls (Fig. 3f). Taken together, these data indicate that *mito-APOBEC1*-induced mutations compromise OXPHOS assembly and respiration likely by the cumulative impact on mitochondrial-encoded OXPHOS components.

**Analysis of the mtDNA mutation profiles.** An unexpected finding from our work concerns the similar mutation loads found in maternal *tam*$^{exo-}$ and *mito-APOBEC1* flies, but the strikingly

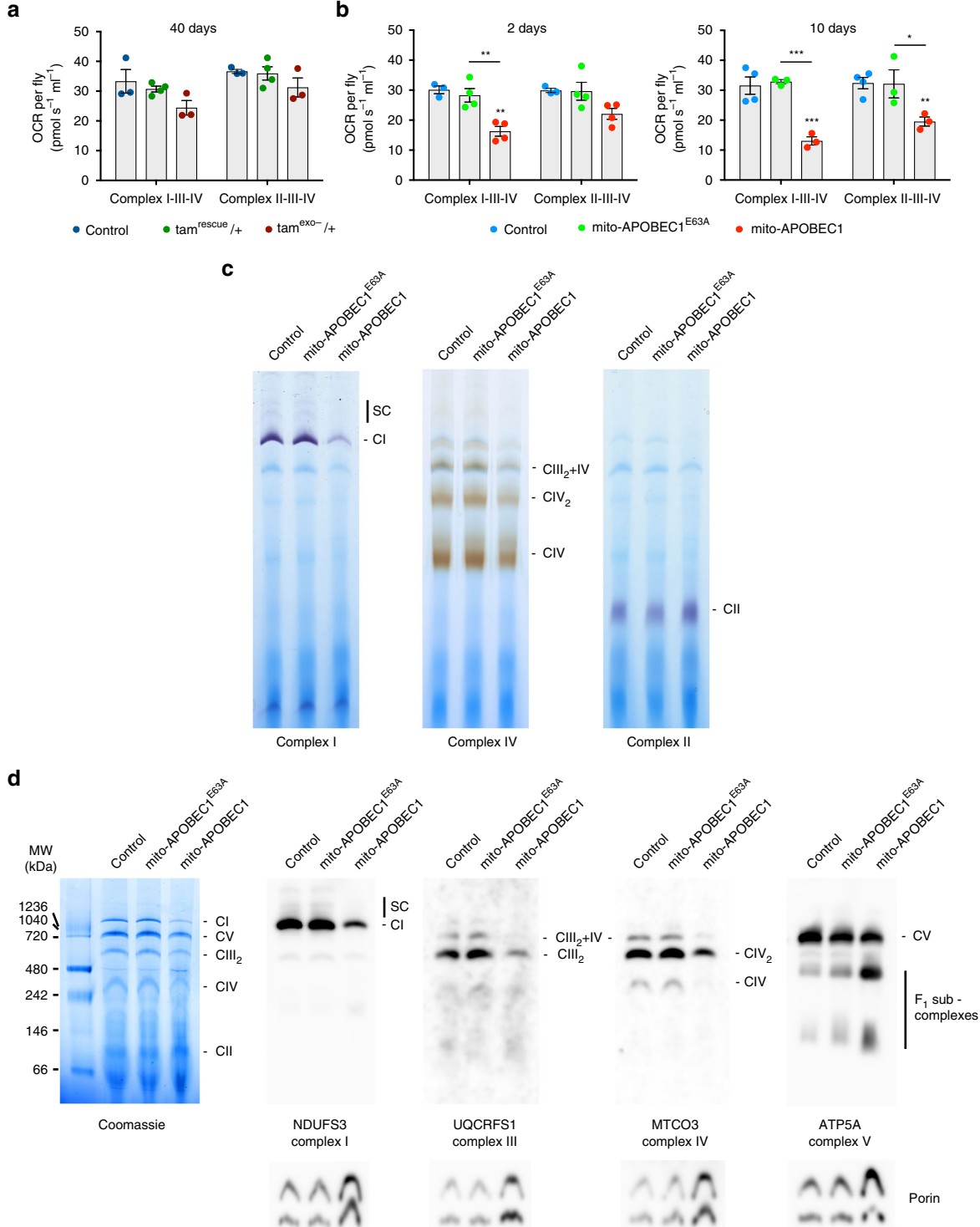

**Fig. 5** Mitochondrial respiration is compromised in *mito-APOBEC1* but not *tam*exo− flies. Respiratory activity was measured in **a** maternally inherited *tam*exo− and **b** *mito-APOBEC1* flies and respective controls by Complex I- or Complex II-linked oxygen consumption rates (OCR) in isolated mitochondria from flies at the indicated age. Values were normalised per fly. Charts show mean ± SEM, $n = 3$ biologically independent samples. Statistical analysis used one-way ANOVA with Sidak's multiple comparisons test; *$P < 0.05$, **$P < 0.01$, ***$P < 0.001$. All other comparisons are non-significant (NS). Blue Native-PAGE followed by in-gel activity assay (**c**) or immunoblotting (**d**) of mitochondrial-enriched samples from *mito-APOBEC1* flies. **c** Gels were incubated for Complex-I or Complex-II activities (violet), or Complex-IV activity (brown). Equal amounts of total protein were loaded per lane. **d** Blue-Native gels were Coomassie stained or used for immunoblotting against NDUFS3 (*Dm* ND-30; Complex-I), UQCRFS1/Rieske (*Dm* RFeSP; Complex-III), MTCO3 (*Dm* mt: CoIII; Complex-IV) and ATP5A (*Dm* Blw; Complex-V). Anti-Porin was used as a loading control. CI, Complex-I; CII, Complex-II; CIII, Complex-III; CIII$_2$, Complex-III dimer; CIV, Complex-IV; CIV$_2$, Complex-IV dimer; CV, Complex-V; SC, super-complexes. Source data are provided as a Source Data file

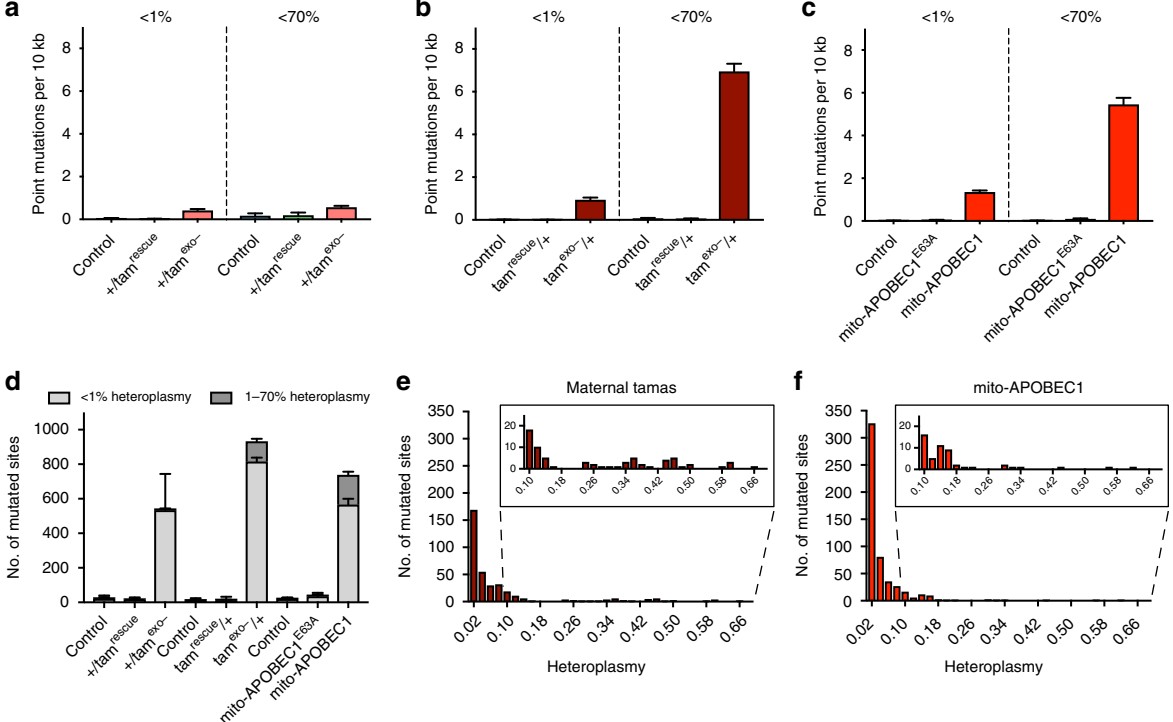

**Fig. 6** High levels of heteroplasmy occurs in maternal *tam*^exo− and *mito-APOBEC1*. Point mutation load of low heteroplasmy events (<1%) compared to all observed mutations (<70% heteroplasmy) found in 10-day-old **a** paternally inherited *tam*^exo−, **b** maternally inherited *tam*^exo− and **c** *mito-APOBEC1* mutator flies and respective controls. Mutation loads for total mutations (<70% heteroplasmy) are the same as shown in Fig. 1a, and reproduced here for comparison. **d** Number of sites that show low (<1%) or moderate (1–70%) heteroplasmy. Charts show mean ± SD, *n* = 3 animals. **e**, **f** Frequency distribution of >1% heteroplasmic sites in **e** maternally inherited *tam*^exo− and **f** *mito-APOBEC1* flies. Each chart shows the frequency distribution of the number of mutated sites with heteroplasmy bin width = 0.02, centred as indicated. Sites with heteroplasmy spanning 9 to 69% are zoomed. Source data are provided as a Source Data file

different organismal consequences of those mutations. This suggests that the amount of mtDNA mutations alone is not sufficient to explain the difference in pathogenicity. Therefore, we undertook a deeper analysis of the mtDNA mutation profile observed in the two mutator systems.

We first assessed the degree to which the observed mutations may have undergone clonal expansion. Mutations that occur at less than 1% heteroplasmy are generally considered to represent recently acquired or de novo mutational events, while mutations present at higher than 1% heteroplasmy are thought to have expanded clonally through replication of an initial event. We compared the overall mutation load, up to 70% heteroplasmy, with that found at less than 1% heteroplasmy (Fig. 6a–c, and Supplementary Data 1). Paternally inherited *tam*^exo− flies showed very little difference in mutation load between <1% and <70% heteroplasmy (Fig. 6a), consistent with these mutations being recently acquired. Indeed, only a few sites in paternally inherited *tam*^exo− flies showed >1% heteroplasmy (Fig. 6d, Supplementary Fig. 1b). In contrast, a substantial proportion of mutations in *mito-APOBEC1* and maternally inherited *tam*^exo− occurred at higher than 1% heteroplasmy (Fig. 6b, c, and Supplementary Fig. 1b). Notably, 23% of all mutated sites in *mito-APOBEC1* (Fig. 6d) displayed a moderate level of heteroplasmy (mainly between 1–10% heteroplasmy, Fig. 6f), while in *tam*^exo−/+ a smaller proportion of clonally mutated sites (13% of all mutated sites, Fig. 6d) distributed as a few moderately heteroplasmic positions together with a handful of high clonally (>30% heteroplasmy) mutated sites (Fig. 6e).

Since the preceding mutation analyses were performed on 10-day-old flies, we wanted to determine the extent to which these

mutations were present in early adulthood, in 2-day-old flies (Supplementary Data 3). Mutation loads were similar between 2- and 10-day-old flies across all genotypes tested (Supplementary Fig. 5a). However, while a similar number of sites were mutated in *mito-APOBEC1* flies at 2 and 10 days, the number of mutated sites approximately doubled in both *tam*^exo− conditions (Supplementary Fig. 5b). Therefore, during this short time frame, *tam*^exo− was able to introduce mutations at additional genomic positions, consistent with reports that mtDNA replication indeed occurs in adult fly tissues[27]. In contrast, the lack of increase in *mito-APOBEC1*-targeted sites suggests the mutable positions may have already been saturated by 2 days. Consistent with this, comparing the heteroplasmy profile of 2- and 10-day-old *tam*^exo− flies showed no appreciable increase in the number of clonally mutated sites (1–70% heteroplasmy) while the number of sites with de novo mutations (<1% heteroplasmy) specifically increased (Supplementary Fig. 5b). It should be noted that the additional mutations at <1% heteroplasmy in 10-day-old flies are numerically very small compared to the large number of mutations present at higher (1–70%) heteroplasmy, accounting for why the overall point mutation load does not significantly increase with age. As above, the heteroplasmy profile of *mito-APOBEC1*-targeted sites did not change between 2 and 10 days (Supplementary Fig. 5b).

Being unable to ascribe the divergent organismal outcome of *tam*^exo− and *mito-APOBEC1* flies to difference in their heteroplasmy profile, we next considered whether the topography of the mutations may offer some explanation as to the different pathogenicity. In all three mutator models, point mutations were widely distributed across the mtDNA genome (Supplementary

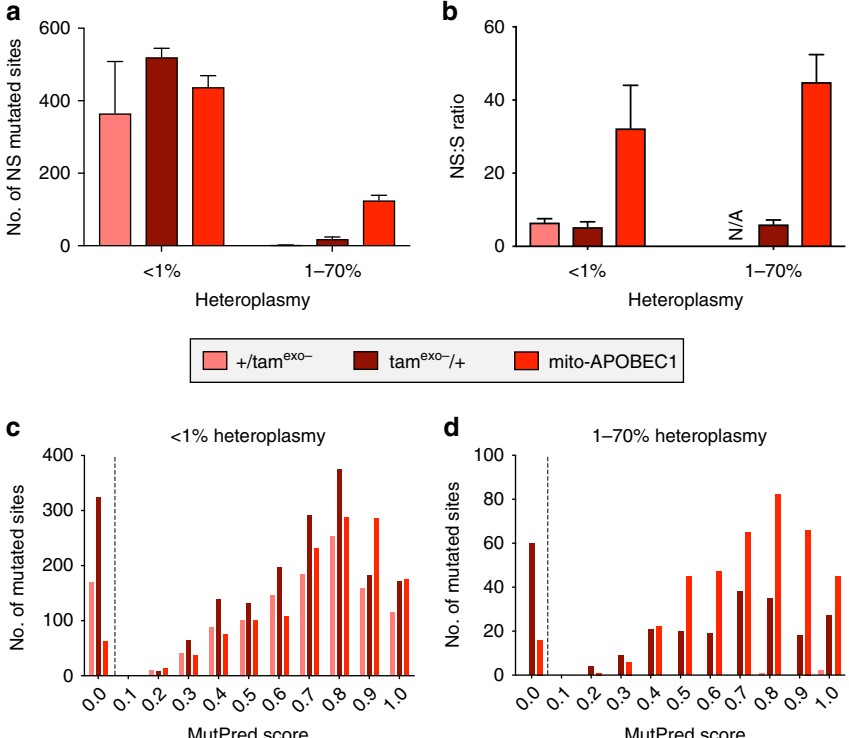

**Fig. 7** *mito-APOBEC1* causes high levels of non-synonymous pathogenic mutations. **a** Number of genomic sites presenting non-synonymous (NS) substitutions, at <1% or 1–70% heteroplasmy, in the indicated genotypes. **b** Non-synonymous:synonymous (NS:S) ratio of all substitutions recovered in protein-coding genes, separated by heteroplasmy level as previously indicated [Note: Only one non-synonymous substitution was found at >1% heteroplasmy in paternally inherited *tam*^exo− and 0 synonymous mutations; therefore, the ratio is indicated as N/A]. Charts show mean ± SD, *n* = 3 animals. **c, d** Frequency distribution of MutPred scores, bin width = 0.05, for each mutation found in the indicated genotypes analysed by **c** <1% or **d** 1–70% heteroplasmy. (*n* = 3 animals, data from each animal were sum by genotype). A dotted line separates the synonymous (MutPred score = 0) from non-synonymous mutations. Source data are provided as a Source Data file

Fig. 1), with no particular bias towards any particular region or genomic feature; protein coding, tRNA, rRNA or non-coding (Supplementary Fig. 2b). Focussing on the protein-coding genes, we calculated the ratio of non-synonymous to synonymous (NS: S) mutations found in each genotype (Supplementary Fig. 6a). Non-synonymous mutations in either of the *tam*^exo− mutator lines essentially exist as recent or de novo events (<1% heteroplasmy, Fig. 7a), and the NS:S ratio in these lines is low (Fig. 7b) and similar to the ratio found in controls (Supplementary Fig. 6a). In contrast, *mito-APOBEC1* produced a high number of NS mutations, particularly at sites with >1% heteroplasmy (Fig. 7a), resulting in a dramatic increase in the NS:S ratio (Fig. 7b).

Several characteristics of the mutation profile likely contribute to this difference between the *tam*^exo− and *mito-APOBEC1* mutators. First, considering that *mito-APOBEC1* exclusively targets C:G nucleotides (Supplementary Fig. 2a), it is notable that C:Gs are predominantly found in protein-coding regions (Supplementary Fig. 6b), particularly in the critical first and second codon-base positions (Supplementary Fig. 6c). Indeed, *mito-APOBEC1* disproportionately (~98% of all recovered mutations) mutates the first or second codon-base position (Supplementary Fig. 6d). Consistent with *tam*^exo− creating a greater diversity of substitutions (Supplementary Fig. 2a), a substantial portion (21%) of *tam*^exo−-induced mutations affects nucleotides at the third position in the codon (Supplementary Fig. 6d), sites that tend to be degenerate (mutations lead to synonymous substitutions). Second, *mito-APOBEC1* is highly selective for targeting deamination of cytidine in a TC*n* context,

particularly in the TCT trinucleotide sequence (Supplementary Fig. 7), as previously reported[32]. Therefore, *mito-APOBEC1*-induced mutations are restricted to fewer sites in the genome, which may expand to higher heteroplasmy due to multiple, independent targeting events. In contrast, similar to the codon position analysis, *tam*^exo− mutators show much less selectivity for the surrounding nucleotide context (Supplementary Fig. 7). Third, a significant proportion of *tam*^exo−-induced mutations are T>C substitutions, which are much more likely to result in a synonymous change (Supplementary Fig. 6e). Taken together, these data demonstrate that *mito-APOBEC1* induces predominantly non-synonymous mutations.

As a further estimate of the pathogenicity of the mutations that each model introduces at protein-coding genes, we calculated the MutPred pathogenicity score (see 'Methods' section) for every mutated site recovered in each of our mutator genotypes at 10 days old. All three mutators show a similar MutPred score distribution when considering sites at <1% heteroplasmy, with either of *tam*^exo− mutators having a substantial abundance of synonymous (0-scored) sites (Fig. 7c). However, higher heteroplasmy sites (1–70%) in *mito-APOBEC1* show a strong preponderance for elevated MutPred scores (Fig. 7d). As for <1% heteroplasmy sites, a substantial proportion of the 1–70% heteroplasmy positions in *tam*^exo−/+ are synonymous and therefore non-pathogenic (Fig. 7d).

Taken together, these analyses indicate that the high selectivity of *mito-APOBEC1* for C:G>T:A transitions generates a remarkably high proportion of non-synonymous and potentially pathogenic mutations which, despite achieving only moderate

heteroplasmy, confer a strongly detrimental effect on mitochondria. The different pathogenicity of the mutations introduced by *mito-APOBEC1* or *tamas* mutators offers a plausible explanation for the striking differences on organismal lifespan and vitality between the two models.

## Discussion

The accumulation of mtDNA mutations has been implicated in various age-related pathologies and in the ageing process itself[5,6]. The *POLG*-based mutator systems, which display a striking progeroid syndrome in mice[9,10], have emerged as the predominant model for obtaining high levels of stochastic mtDNA mutations. However, the mutational heterogeneity that arises in this model, which includes point mutations as well as small indels, large deletions, and mtDNA depletion, has led to considerable debate about the pathogenic entity and whether point mutations are actually driving ageing[11–14]. Recently described fly and worm *POLG* mtDNA mutators, which offer an attractive system for their powerful genetic approaches, also show evidence of this mutational heterogeneity[15,16,33]. However, while the first fly model equivalent to the *POLG*-mutator was homozygous lethal[15], a similar model recently reported is not lethal[16], raising uncertainty about the cause of the lethality. To establish a 'cleaner' mutator system we have developed an alternative mutator model, *mito-APOBEC1*, which we showed is a potent mtDNA mutagen that exclusively induces high levels of C:G > T:A transition mutations, in the absence of indels or mtDNA depletion.

Because we believe the level of heteroplasmy will contribute to the pathogenicity of a mutation profile, the mutation loads calculated here reflect the overall amount of mutations we recovered, including multiple events at a same site. This clonality is generally assumed to be due to clonal propagation of an initial mutational event via replication. However, it is also possible that the same mutational event can independently occur in different molecules, in an iterative mechanism that is independent of mtDNA replication. This is particularly relevant for our *mito-APOBEC1* model which has a very high degree of sequence selectivity and is not directly linked to replication unlike a polymerase-based (*tam*exo−) model.

Comparing the POLG/*tamas* and *mito-APOBEC1* model systems, we find that although a similarly high mtDNA mutation load can arise in both *mito-APOBEC1* and maternally inherited *tam*exo− flies, they cause dramatically different effects on organismal fitness. While mutations generated by *mito-APOBEC1* lead to profound shortening of lifespan and loss of vitality (Fig. 4e, f), associated with a broad spectrum of mitochondrial impairments (Fig. 5b–d), *tam*exo− does not provoke any major defects either at the organelle or organismal level (Figs. 4a–d and 5a).

The in-depth mutation analysis method we used allowed us to gain a detailed view of the mutation profiles in these models. First, there is a striking difference (> 10-fold) in the overall mutation load when *tam*exo− is inherited paternally versus maternally (Fig. 1a). This is accounted for by a number of clonally expanded mutations (Fig. 6a, b, d, e) which likely arise through replication of germline transmitted mutations or very early mutagenesis by maternally deposited Tamexo− protein, which is abundant enough to allow *tam* mutants to development to the third instar stage[34]. The increase in de novo but not clonally mutated sites between 2- and 10-day-old *tam*exo− flies (Supplementary Fig. 5b) further supports that mtDNA continues to undergo replication in adults[27], but that clonal expansion events likely occur only in development, probably during the peak of mtDNA replication part-way through embryogenesis[28]. Interestingly, *mito-APOBEC1*, which is only expressed in the zygote

and not maternally transmitted, also produces a substantial amount of highly heteroplasmic sites (Fig. 6c), in stark contrast to paternally inherited (zygotically expressed) *tam*exo− (Fig. 6a). *mito-APOBEC1*-induced mutations could expand to high heteroplasmy through replication but also via iterative targeting of cytidines on different mtDNA molecules. Further work will be needed to determine the mechanistic basis for this phenomenon, but we hypothesise that, since APOBEC1 is known to bind single-stranded DNA[20], mutations may occur during transcription in addition to during replication, as it has previously been suggested[32]. Remarkably, the mutation load, number of mutated sites and degree of heteroplasmy did not change in the *mito-APOBEC1* flies between 2 and 10 days (Supplementary Fig. 5). We hypothesise that *mito-APOBEC1* is such an efficient mutator that the mutable sites compatible with life are saturated early in development. Further insight into these mechanisms could be gained from analysing the changes in mutation profile during developmental stages in these models. Similarly, it will also be informative to explore which, if any, additional changes occur during a more substantial period of ageing.

Eliminating differences in heteroplasmy levels as a mechanistic explanation for the differing organismal phenotypes between *tam*exo− and *mito-APOBEC1*, we focused our analyses on the mutation profile. While both models show a predisposition to create C:G>T:A mutations (Fig. 1d and Supplementary Fig. 2a), *mito-APOBEC1* leads to a higher proportion of potentially damaging mutations and non-synonymous substitutions (Fig. 7). The increased pathogenicity of the *mito-APOBEC1* spectrum is likely caused by its exclusive targeting of C:G nucleotides (Supplementary Fig. 2a), particularly in a TCn context (Supplementary Fig. 7), while *tam*exo− has a more variable spectrum of substitutions, including transversions (Supplementary Fig. 2a), and a less stringent neighbouring sequence requirement (Supplementary Fig. 7). Notably, *Drosophila* mtDNA has evolved to have an unusually low C:G content (Supplementary Fig. 6b), likely retaining them predominantly at critical sites (Supplementary Fig. 6c); thus, C:G mutations are presumably more likely to be damaging[31,35]. Consequently, we surmise that mtDNA mutation type (quality) rather than quantity is a critical factor in impacting organismal fitness. One limitation of our work is that we have restricted our sequence analyses predominantly to protein-coding sequence. Future work will determine whether the context specificity of *mito-APOBEC1* also leads to an enrichment of pathogenic mutations within tRNA and rRNA sequences, and how these may impact organismal fitness and ageing.

The fact that *mito-APOBEC1* flies show such detrimental phenotypes but only modest levels of heteroplasmy is also intriguing. Typically, for mitochondrial diseases resulting from mtDNA mutations, heteroplasmy levels must rise above a certain threshold (>60–90%), to cause mitochondrial functional impairment[36,37]. This phenomenon is also replicated in a number of *Drosophila* strains with homoplasmic mtDNA mutations that present strong mitochondrial and neuromuscular defects[38–40]. However, a threshold level can be reached not only by one specific mutation but by the sum of a number of different mutations[14]. Our results from *mito-APOBEC1* flies support this view. Mutations can produce cumulative defects on the same protein or multi-protein complexes that limit the possibility of complementation. Alternatively, mutations can arise locally to very high heteroplasmy and create a mosaic of affected cells within a tissue. Current sequencing technologies, however, cannot distinguish between these scenarios and would appear as apparent lower level heteroplasmy on aggregate. Such a death-by-a-thousand-cuts model could help explaining how the variety of mtDNA mutations frequently observed in ageing and in a number of neurodegenerative diseases could have a phenotypic

effect without ever reaching the high level of heteroplasmy typically associated with mitochondrial diseases[25,41–43].

Finally, we posit that the *mito-APOBEC1* model circumvents several limitations of the existing *POLG*-based mutator models and offers several advantages. First, it allows us to specifically address the impact of point mutations, in the absence of indels or mtDNA depletion. Second, the stronger phenotypes induced in this model provide an opportunity to screen for genetic modifiers to identify factors influencing the pathogenicity of mtDNA mutations. Third, as *mito-APOBEC1* is based on the classic GAL4/UAS transgenic approach this opens it up to the full power and versatility of *Drosophila* genetics, providing an unparalleled toolbox for a range of spatial and temporal manipulation, such as tissue-specific or life course-specific expression. Beyond *Drosophila*, this study acts as a proof-of-principle to extend the inducible *mito-APOBEC1* model system to other organisms where it would provide analogous advantages over existing mutator models to further decipher the pathophysiology of mtDNA mutations.

## Methods

**Drosophila stocks and husbandry**. Flies were raised under standard conditions in a humidified, temperature-controlled incubator with a 12 h:12 h light:dark cycle at 25 °C, on food consisting of agar, cornmeal, molasses, propionic acid and yeast. Transgene expression was induced using the ubiquitous *da-GAL4* driver. The following strains were obtained from the Bloomington *Drosophila* Stock Center (Bloomington Drosophila Stock Center, RRID:SCR_006457): *w1118* (RRID: BDSC_6326), *da-GAL4* (RRID:BDSC_55850), *UAS-mito-HA-GFP* (RRID: BDSC_8443). *wDah* (*w*‾ [Dahomey, Wolbachia-free]), *tamexo−* and *tamrescue* flies were a gift from N-G. Larsson (Max Planck Institute for Ageing, Cologne). Unless otherwise stated, all experiments were conducted using male flies.

## Genotypes used in this study

| Group | Label | Genotype |
| --- | --- | --- |
| Paternal tamas | control | *w1118*; + / + |
| | +/*tamrescue* | *w1118*; + /TI{TI}*tam*Rescue |
| | +/*tamexo−* | *w1118*; +/ TI{TI}*tam*D263A |
| Maternal tamas | control | *wDah*; + / + |
| | *tamrescue* /+ | *wDah*; TI{TI}*tam*Rescue/ + |
| | *tamexo−*/+ | *wDah*; TI{TI}*tam*D263A/+ |
| mito-APOBEC1 | control | *w1118*; P{UAS-mito-HA-GFP}/+; P{da-GAL4}/+ |
| | *mito-APOBEC1*E63A | *w1118*; P{UAS-mito-HA-APOBEC1*E63A*}/+; P{da-GAL4}/+ |
| | *mito-APOBEC1* | *w1118*; P{UAS-mito-HA-APOBEC1}/+; P{da-GAL4}/+ |

**Generation of APOBEC1 transgenic lines**. *mito-APOBEC1* fusion constructs were made by fusing the MTS of human ATP synthase subunit F1β (ATP5F1B), a haemagglutinin (HA) epitope and a flexible linker sequence 5′ to the rat APOBEC1 cDNA which lacked the initial ATG and stop codon. A nuclear exporting sequence (NES) was inserted 3′ to APOBEC1. The Glu63Ala (APOBEC1*E63A*) catalytically dead mutant was created by changing the codon GAA to GCT at amino acid position 63 of the cDNA by site-directed mutagenesis. Mitochondria-excluded, cytoplasmic versions of APOBEC1 constructs (*cyto-APOBEC1*) were generated by removing the MTS sequence from the *mito-APOBEC1* transgenes. Engineered sequences were cloned into the *pUASTattB* vector and injected in *Drosophila* embryos for phiC31-mediated transgenesis at the attP40 site (BestGene, RRID: SCR_012605).

**DNA isolation for sequencing**. Brains from individual male flies of the indicated age were dissected in PBS, flash frozen in dry ice, and stored at −80 °C. Total DNA was isolated from individual brains using the QIAamp DNA Micro isolation kit following the manufacturer's instructions (QIAGEN, RRID:SCR_008539). DNA yield for a single brain typically ranged between 20–30 ng.

**Duplex sequencing and mutation calling**. DNA was prepared for Duplex sequencing using a previously described protocol[44] with several modifications. Briefly, ~20 ng of total DNA was sonicated in 60 µL of nuclease-free ddH2O using a Covaris AFA system with a duty cycle of 10%, intensity of 5, cycles/burst 100, time

20 s × 5, temperature of 4 °C. After sonication, each sample was subjected to end-repair and 3′-dA-tailing using the NEBNext Ultra End-repair/dA-tailing kit (New England Biolabs, RRID:SCR_013517) according to the manufacturer's instructions. Each sample was then ligated with 2 µL of 15 µM Duplex Sequencing adapters, using the NEBNext Ultra Ligation kit (New England Biolabs, RRID:SCR_013517) according to the manufacturer's instructions. Duplex Sequencing adapters used in collecting data for analysing mutation-selection were chemically synthesized as a collaborative effort with Integrated DNA Technologies to develop a prototype synthesis method. Each sample was then cleaned of excess adapters using Agen-Court AmpureXP magnetic beads, and PCR amplified[44]. The mitochondrial DNA was isolated by targeted DNA capture using IDT xGen Lockdown probes (Integrated DNA Technologies OligoAnalyzer, RRID:SCR_001363) specific for non-repeat regions of the *Drosophila* mitochondrial genome, using the manufacturer's instructions.

The captured DNA samples were sequenced on an Illumina NextSeq500 using 150 bp paired-end sequencing. The resulting reads were aligned against the *Drosophila* genome (BDGP Release 6 + ISO1 MT/dm6) using the Burrows-Wheeler Aligner (BWA, RRID:SCR_010910) and Samtools[45] (SAMTOOLS, RRID: SCR_002105) coupled with a custom software workflow[44]. Reads not uniquely mapping to the mitochondrial genome were excluded from further analysis. The non-coding A+T region [ChRM:14917-19524] has a repetitive high A:T content (~95%) which confounds sequence capture and mapping. The majority of A+T region is therefore not included in the calculation of mutation load (see Supplementary Data files for full details). After processing, we called de novo mutations by using a cut-off that excluded variants occurring at > 1% heteroplasmy, while *global* mutation loads included all mutations with heteroplasmy <70% at each position.

**mtDNA mutation analysis**. For each animal, point mutation or indel loads were calculated as the total number of mutations, insertions or deletions per total number of nucleotides sequenced, considering the whole or a subset of the genome, as relevant to each analysis. Full details of sequencing results and mutations per animal are shown in Supplementary Data files. For heteroplasmy distribution analysis, results from individual flies per genotype were aggregated.

The trinucleotide context of mutations was calculated using previously published pipelines on the aggregated sequence data from flies of the same genotype[16,46]. Each mutation was characterized according to both the identity of the mutation as well as the nucleotides 3′ and 5′ of the mutation site, and binned into one of the 96 total possible mutation types. To calculate the load of mutations in each trinucleotide context, we first determined the total genome-wide post-processing duplex depth for each of the 16 total trinucleotide contexts. We then calculated the trinucleotide mutation load for each of the 96 mutation types by dividing the total number of mutations for each mutation type by its post-processing duplex depth.

Calculations for the distribution of mutation positions, trinucleotide context and mutation loads throughout the genome were calculated using Microsoft Excel for Mac v.16 (Microsoft Excel, RRID:SCR_016137) and scripts developed in Python 2.7 or Python 3.6 (Python Programming Language, RRID:SCR_008394). Graphs were generated using GraphPad Prism software (GraphPad Prism, RRID: SCR_002798) or R software (R Project for Statistical Computing, RRID: SCR_001905).

The MutPred software (MutPred, RRID:SCR_010778)[47] was used to calculate pathogenicity scores for all non-synonymous (NS) variants detected in mutator flies. In addition, we assigned a score of 0 to all non-synonymous changes, and a score of 1 to nonsense mutations or mutations changing a stop codon to coding. MutPred scores for all detected single nucleotide changes were aggregated by genotype and plotted as a frequency distribution.

**mtDNA copy number**. Total DNA was extracted from 10–20 male flies using the DNeasy Blood and Tissue kit (Qiagen) and following manufacturer's instructions. Quantification of mtDNA was performed in triplicate by multiplex TaqMan qPCR amplification of the mitochondrial genes mt:CoI and mt:lrRNA, and the nuclear gene RpL32 as normalizing reference. The following primers were used: (mt:CoI) 5′-TTCTACCTCCTGCTCTTTCTTTAC and 5′-CAGCGGATAGAGGTGGA-TAAAC, probe 5′-FAM-AATGGAGCTGGGACAGGATGAACT-TAMRA; (mt: lrRNA) 5′-AGATAGAAACCAACCTGGCTTAC and 5′-TTGGGTGTAGCCGTTCAAAT, probe 5′-FAM-ACCGGTTTGAACTCAGA TCATGTAAGA-TAMRA; (RpL32) 5′-CACCGGAAACTCAATGGATACT and 5′-CACACAAGGTGTCCCACTAAT, probe 5′-FAM-CCAAGAAGC TAGCCCAACCTGGTT-TAMRA. PCR reactions were performed according to standard conditions for TaqMan (Applied Biosystems, RRID:SCR_005039): 50 °C for 2 min; 95 °C for 10 min; 40 cycles at 95 °C 15 s, 60 °C 1 min. The expression of mtDNA copy number relative to nuclear DNA was determined using the $2^{-\Delta\Delta CT}$ method. The relative quantification was corrected for PCR efficiency of each primer pair.

**Southern blot**. Total DNA was extracted from 150–180 7-day old flies using phenol:chloroform following the protocol in the VDRC website (Vienna Drosophila Stock Center, RRID:SCR_013805) (https://stockcenter.vdrc.at/images/

downloads/GoodQualityGenomicDNA.pdf). 5 ug of DNA was digested with *Pst*I (New England Biolabs, RRID:SCR_013517) according to manufacturer's instructions, and resolved on a 0.55% agarose gel. Blotting, labelling of probe DNA and membrane hybridisation were performed using standard procedures[48]. Membranes were exposed to a storage phosphor screen for 24–96 h before imaging with an Amersham Typhoon RGB scanner and processing with ImageQuant software (ImageQuant, RRID:SCR_014246). Probe primer sequences were as follows: (mt: CoII—3276–3840) 5′-AACTATTTTACCAGCAATTATTTTACT and 5′-CAGTC ATCTAATGAAGAGTTATTTCTA; (18S rDNA) 5′-CGATGCCAGCTAGCAA TTGGGTGTAG and 5′-CTACACCCAATTGCTAGCTGG.

**Long-range PCR**. Total DNA was extracted from 10–20 male flies using the DNeasy Blood and Tissue kit (Qiagen) and following manufacturer's instructions. PCR amplification was performed using PrimeSTAR GXL DNA Polymerase under manufacturer's conditions (Clontech, RRID:SCR_004423), with 1 µM of primers and the following amplification conditions: 94 °C, 1 min; 98 °C, 30 s; 68 °C, 13 min (30 cycles); 72 °C, 10 min. Primers used (14.2F: 5′-GCCGCTCCTTTCCATT TTTGATTTCC and 14.2R: 5′-TGCCAGCAGTCGCGGTTATACCA) amplify a product encompassing almost the complete mtDNA molecule. The PCR products were then visualized after electrophoresis on 0.8% agarose and 2X Invitrogen SYBR Safe DNA Gel Stain (Thermo Fisher Scientific, RRID:SCR_008452) and 1 kb DNA Ladder (New England Biolabs, RRID:SCR_013517).

**qRT-PCR of mtDNA transcripts**. Total RNA was extracted from 10 7-days old male flies using the Direct-zol MiniPrep kit (Zymo Research, RRID:SCR_008968). Genomic DNA contamination was removed from RNA using Turbo DNAse (Ambion Inc, RRID:SCR_008406), according to manufacturer's protocol. cDNAs were generated from 80 ng of total RNA using the Maxima H Minus cDNA Synthesis Master Mix with dsDNAse (Thermo Fisher Scientific, RRID: SCR_008452), following manufacturer's instructions. To ensure that RNA had no genomic DNA contamination, a control reaction was included in which no reverse transcription was carried out. Reactions were carried out using a QuantStudio 3 Real-Time PCR Systems (Thermo Fisher Scientific, RRID:SCR_008452) with Maxima SYBR Green/ROX qPCR Master Mix (Thermo Fisher Scientific, RRID: SCR_008452). Primers were as follows: (mt:CoI) 5′-CAGGATGAACTGTTT ATCCACCTTT and 5′-AATCCCTGCTAAATGTAGAGAAAAAATAG; (mt: ND3) 5′-AAAAAGCTTTAATCGACCGAGA and 5′-CGTAAAGAA AATGGTAATCGAGATG; (mt:CytB) 5′-AAATTTATTGGGGAGACCCTGATA AC and 5′-GGAATAGATCGTAAAATAGCATAAGCA; (mt:ND4) 5′-AACCC AGAAGAACATAAACCA and 5′-TGCTTATTCATCTGTTGCTCA; (mt:lrRNA) 5′-ACCTGGCTTACACCGGTTT and 5′-GGGTGTAGCCGTTCAAATTT; (RpL32) 5′-AAACGCGGTTCTGCATGAG and 5′-GCCGCTTCAAGGGAC AGTATCTG; (Tub84b) 5′-TGGGCCCGTCTGGACCACAA and 5′-TCG CCGTCACCGGAGTCCAT. Primers' specificity was assessed by melting curve profile and their efficiency ranged from 0.92 to 0.99. Mitochondrial transcripts levels were normalised to a geometric mean of both Rpl32 and Tub84b reference genes, using the comparative Ct method. The relative quantification was corrected for PCR efficiency of each primer's couple.

**Locomotor and lifespan assays**. The startle induced negative geotaxis (climbing) assay was performed using a counter-current apparatus. Briefly, 20–23 males were placed into the first chamber, tapped to the bottom, and given 10 s to climb a 10 cm distance. This procedure was repeated five times (five chambers), and the number of flies that has remained into each chamber counted. The weighted performance of several group of flies for each genotype was normalized to the maximum possible score and expressed as *Climbing index*[49].

For lifespan experiments, flies were grown under identical conditions at low-density. Progeny were collected under very light anaesthesia and kept in tubes of approximately 25 males each. Flies were transferred every 2–3 days to fresh medium and the number of dead flies recorded. Percent survival was calculated at the end of the experiment after correcting for any accidental loss.

**Immunofluorescence experiments**. For APOBEC1 immunostaining, larval epidermis was dissected in PBS and fixed in 4% formaldehyde for 30 min at RT, followed by permeabilization in 0.3% Triton X-100 for 30 min and blocking with 0.3% Triton X-100, 1% BSA in PBS for 1 h at RT. Tissues were incubated with anti-HA and anti-ATP5A antibodies diluted in blocking buffer overnight at 4 °C, and with the appropriate fluorescent secondary antibodies for 2 h at RT. Samples were washed several times in PBS and mounted on slides using Prolong Diamond Antifade mounting medium (Thermo Fisher Scientific, RRID:SCR_008452).

**Microscopy**. Confocal imaging was conducted using a Zeiss LSM 880 microscope (Carl Zeiss MicroImaging) equipped with Nikon Plan-Apochromat 63 × /1.4 NA oil immersion objective. Images were prepared using Fiji software (Fiji, RRID: SCR_002285).

**Respirometry**. Mitochondrial respiration was assayed at 30 °C by high-resolution respirometry using a Oxygraph-2k high-resolution respirometer (OROBOROS

Instruments) using a chamber volume set to 2 mL. Calibration with the air-saturated medium was performed daily. Data acquisition and analysis were carried out using Datlab software (OROBOROS Instruments). Five flies per genotype were homogenised in Respiration Buffer [120 mM sucrose, 50 mM KCl, 20 mM Tris–HCl, 4 mM KH$_2$PO$_4$, 2 mM MgCl$_2$, and 1 mM EGTA, 1 g L$^{-1}$ fatty acid-free BSA, pH 7.2]. For coupled (state 3) assays, complex I-linked respiration was measured at saturating concentrations of malate (2 mM), glutamate (10 mM) and adenosine diphosphate (ADP, 2.5 mM). Complex II-linked respiration was assayed in Respiration Buffer supplemented with 0.15 µM rotenone, 10 mM succinate and 2.5 mM ADP. The addition of proline to the respiration buffer can increase respiration rate in insect samples but was not included here. Respiration was expressed as oxygen consumed per fly. Flies' weight was equal in all genotypes tested.

**Biochemical assays**. Mito-APOBEC1 flies and controls were aged to 10–12 days before collection for mitochondria isolation. 1 mL of flies per genotype were homogenised in 2 mL of STE buffer [250 mM Sucrose, 5 mM Tris, 2 mM EGTA, pH 7.4] + 1% BSA with several strokes in a Teflon-glass homogeniser at 700 rpm. Nuclei and fly body debris were pelleted by two centrifugation steps at 1000 × *g* for 5 min, 4 °C and the supernatant centrifuged for 10 min at 3000 × *g*, 4 °C. After a wash in 5 mL STE buffer + 1% BSA, the mitochondrial pellet was resuspended in STE buffer (without BSA) and centrifuged at 7000 × *g*, 4 °C for 10 min. Mitochondria pellet was resuspended in 1 mL of STE buffer for protein quantification using the Pierce BCA method (Thermo Fisher Scientific, RRID:SCR_008452). Sample preparation and Blue-Native PAGE were performed as follows: 800 µg of pelleted mitochondria were resuspended in 1.5 M aminocaproic acid, 50 mM Bis-Tris·HCl pH 7 to a final concentration of 10 mg mL$^{-1}$ and solubilised with 4 mg digitonin per mg of protein. After 5 min incubation on ice, mitochondria were centrifuged at maximum speed (20,000 × *g*) at 4 °C for 30 min. Supernatant was mixed with 10 µL of Blue-Native Sample Buffer [750 mM aminocaproic acid, 50 mM Bis-Tris·HCl pH 7, 0.5 mM EDTA, 5% Serva Blue G] and 10 µL of extracts loaded for Blue-Native Gel Electrophoresis on a pre-cast NativePAGE 3–12% Bis-Tris gel (Life Technologies, RRID:SCR_008817). For in-gel complex activity, the BN-PAGE gel was incubated for several hours at room temperature in complex-specific solutions. For complex-I activity: 5 mM Tris–HCl, pH 7.4, 0.1 mg mL$^{-1}$ NADH, and 2.5 mg mL$^{-1}$ NTB (NitroTetrazolium Blue); for complex-II activity: 5 mM Tris–HCl, pH 7.4, 0.2 mM phenazine methasulfate, 20 mM sodium succinate and 2.5 mg mL$^{-1}$ NTB; for complex-IV: 50 mM sodium phosphate buffer, pH 7.4, 0.5 mg mL$^{-1}$ DAB (3,3′-diaminobenzidine tetrahydrochloride), 24 U mL$^{-1}$ catalase, 10 mg mL$^{-1}$ cytochrome c (horse heart < 95% purity), 75 mg mL$^{-1}$ sucrose.

**Subcellular fractionation**. 200–250 adult flies (both males and females) were homogenised in 4 mL of cold 250-STM buffer [250 mM sucrose, 50 mM Tris·HCl pH 7.4, 5 mM MgCl$_2$] freshly supplied with 1 mM DTT, 1 mM PMSF, 25 µg mL$^{-1}$ Spermine and 25 µg mL$^{-1}$ Spermidine. The homogenate was then sieved through a 70 µm cell strainer (Corning BV) to remove gross body parts (legs and wings). An aliquot of the decanted homogenate was collected as total fraction, and the rest centrifuged at 800 × *g* for 15 min at 4 °C using a swing-out rotor. Supernatant was stored on ice for further isolation of mitochondria and cytosolic fraction. Nuclei-containing pellet was washed with lysis buffer and nuclei isolated at the bottom of a 2M-STMDPS [2 M sucrose, 50 mM Tris·HCl pH 7.4, 5 mM MgCl$_2$, 1 mM DTT, 1 mM PMSF, 25 µg mL$^{-1}$ Spermine and 25 µg mL$^{-1}$ Spermidine] cushion after ultracentrifugation at 80,000 × *g* for 30 min at 4 °C (Beckman Coulter, RRID: SCR_008940 ultracentrifuge). The nuclear pellet was lysed with 50 µL of NE buffer [20 mM HEPES pH 7.9, 0.5 M NaCl, 1.5 mM MgCl$_2$, 0.2 mM EDTA, 20% glycerol] for at least 30 min at 4 °C and nuclear fraction stored at −80 °C. Mitochondria were separated from cytosol by centrifugation of the nuclei-free homogenate at 6000 × *g* for 15 min using a swing-out rotor. After a quick wash of the pellet, mitochondria proteins were extracted in 70 µL of HDP buffer [10 mM HEPES pH 7.9, 1 mM DTT, 1 mM PMSF] and stored at −80 °C. Supernatant from the mitochondrial centrifugation was processed to obtain a clean cytosolic fraction by ultracentrifugation at 100,000 × *g* for 1 h at 4 °C (Beckman Coulter, RRID: SCR_008940 ultracentrifuge) (pellet discarded) and stored at −80 °C. Samples from each fraction were quantified using Pierce BCA assay (Thermo Fisher Scientific, RRID:SCR_008452) and 15 µg of total protein lysate were loaded on a 4–20% gel (Bio-Rad Laboratories, RRID:SCR_008426), resolved via SDS-PAGE (Bio-Rad Laboratories, RRID:SCR_008426) and blotted to a nitrocellulose membrane. Histone-H3 immuno-reactivity was used to mark the nuclear fraction, ATP5A was used as a mitochondrial marker and GAPDH as a cytoplasmic marker.

**Immunoblotting**. For APOBEC1 expression analysis, protein samples were isolated from whole adult flies. Flies were homogenized in RIPA lysis buffer [50 mM Tris·HCl, 150 mM NaCl, 1 mM EDTA, 1% Triton X-100, 0.5% SDS] with 1 mM PMSF and protease inhibitor mixture (Roche, RRID:SCR_001326). Protein extracts were quantified using the Pierce BCA assay (Thermo Fisher Scientific, RRID: SCR_008452). Typically, 30 µg of protein were resolved by SDS-PAGE (Bio-Rad Laboratories, RRID:SCR_008426) and transferred onto nitrocellulose membrane using a semi-dry Transblot apparatus (Bio-Rad Laboratories, RRID:SCR_008426) according to the manufacturer's instructions. Membranes were blocked with 5%

skimmed milk in TBST [Tris-buffered saline, 0.1% Tween 20] for 1 h at RT and incubated with primary antibody overnight at 4 °C. After several washes in TBST, appropriate horseradish peroxidase-conjugated secondary antibodies were incubated for 1 h at RT. Detection was achieved with ECL-Plus detection kit (GE Healthcare, RRID:SCR_000004) using a ChemiDoc XRS + molecular imager (Bio-Rad Laboratories, RRID:SCR_008426) and analysed by Image Lab Software (Image Lab Software, RRID:SCR_014210).

For immunoblotting of native OXPHOS complexes, Blue Native gels of mitochondria-enriched samples were transferred onto PVDF membranes via wet transfer (Bio-Rad Laboratories, RRID:SCR_008426) following manufacturer's instructions. Membranes were blocked and incubated with primary and secondary antibodies as described above. Detection was achieved with ECL-Plus detection kit (GE Healthcare, RRID:SCR_000004) using an Amersham Imager 600 imaging device.

**Antibodies and dyes**. For immunofluorescence experiments, the following primary antibodies were used: mouse anti-ATP5A (Abcam Cat# ab14748, RRID: AB_301447; 1:500), rabbit anti-HA (Abcam Cat# ab9110, RRID:AB_307019; 1:500). Secondary antibodies were: anti-rabbit AF647 (Thermo Fisher Scientific Cat# A-21244, RRID:AB_2535812; 1:200), anti-mouse AF488 (Thermo Fisher Scientific Cat# A-11001, RRID:AB_2534069; 1:200).

For immunoblot experiments, the following antibodies were used: mouse anti-Actin (Millipore Cat# MAB1501, RRID:AB_2223041; 1:5,000), mouse anti-APOBEC1 (E-2) (Santa Cruz Biotechnology Cat# sc-166508, RRID:AB_2057252; 1:1,000), mouse anti-ATP5A (Abcam Cat# ab14748, RRID:AB_301447; 1:20,000), mouse anti-GAPDH (GeneTex Cat# GTX627408, RRID:AB_11174761; 1:2,000), rabbit anti-HA (Abcam Cat# ab9110, RRID:AB_307019; 1:2,000), rabbit anti-HistoneH3 (Abcam Cat# ab1791, RRID:AB_302613; 1:1,000), mouse anti-MTCO3 (Abcam Cat# ab110259, RRID:AB_10859925; 1:600), mouse anti-NDUFS3 (Abcam Cat# ab14711, RRID:AB_301429; 1:600), rabbit anti-Porin (Millipore Cat# PC548, RRID:AB_2257155; 1:5,000), mouse anti-Tubulin (Sigma-Aldrich Cat# T6793, RRID:AB_477585, 1:5,000), mouse anti-UQCRFS1 (Abcam Cat# ab14746, RRID: AB_301445; 1:600). Horseradish peroxidase-conjugated secondary antibodies: anti-mouse (Abcam Cat# ab6789, RRID:AB_955439; 1:5,000 to 1:20,000), anti-rabbit (Thermo Fisher Scientific Cat# G-21234, RRID:AB_2536530; 1:3,000 to 1:5,000).

**Statistical analysis**. Data are reported as mean ± SD, mean ± SEM or mean ± 95% CI as indicated in figure legends. For statistical analyses of lifespan experiments, a log-rank (Mantel–Cox) test was used. For behavioural analyses, a Kruskal–Wallis nonparametric test with Dunn's post-hoc correction for multiple comparisons was used. Significance in mtDNA copy number by qPCR and mitochondrial transcripts analyses was determined by a two-way ANOVA test with Tukey's post-hoc correction for multiple comparisons. Significance in Southern blot quantification and Oroboros (respirometry) experiments was determined by a one-way ANOVA test with Sidak's post-hoc correction for multiple comparisons. Significance levels are indicated in the figure legends. Unless specifically indicated, no significant difference was found between a sample and any other sample in the analysis. Statistical analyses were performed using GraphPad Prism 7 software (GraphPad Prism, RRID:SCR_002798).

## Data availability

The source data underlying all Figs. and Supplementary Figs. are provided as a Source Data file. A Reporting Summary for this article is available as a Supplementary Information file. All data that support the findings of this study are available on reasonable request to the corresponding author. The contributing authors declare that all relevant data are included in the paper and its supplementary information files.

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

## Acknowledgements

This work is supported by MRC core funding (MC_UU_00015/4, MC-A070-5PSB0 and MC_UU_00015/6) and ERC Starting grant (DYNAMITO; 309742). S.A. was supported by an MRC Career Development Fellowship. A.G.D. receives support from NIHR Biomedical Research Centre pilot studies (RROI.GAAB). Work was further supported by DOD/CDMRP grant W81XWH-16-1-0579 to S.R.K., National Institute of Neurological Disorders and Stroke (R21NS090073) to L.J.P., and the Genetic Approaches to Aging Training Grant (T32AG000057) to C.L.S. P.F.C. is a Wellcome Trust Principal Research Fellow (212219/Z/18/Z), and a UK NIHR Senior Investigator, who receives support from the Medical Research Council Mitochondrial Biology Unit (MC_UP_1501/2), the Medical Research Council (UK) Centre for Translational Muscle Disease (G0601943), the Evelyn Trust, and the National Institute for Health Research (NIHR) Biomedical Research Centre based at Cambridge University Hospitals NHS Foundation Trust and the University of Cambridge. The views expressed are those of the author(s) and not necessarily those of the NHS, the NIHR or the Department of Health. Stocks were obtained from the Bloomington *Drosophila* Stock Center which is supported by grant NIH P40OD018537. We thank the Whitworth lab for discussions and for critical reading of the manuscript, and Aurelio Reyes for help with the mitochondrial transcription and replication analysis.

## Author contribution

A.J.W. conceived the study and supervised the work with L.J.P. and S.R.K. S.A., A.S.-M., E.F.-V., A.G.-D., J.J.L., R.T., M.J.H., E.K.S., and P.A.G. designed and performed experiments. C.S. and S.A. performed computational analyses on mtDNA mutations. A.G.-D. and P.F.C. assisted with mtDNA data interpretation. T.J.N. and M.M. conceived the use of mito-APOBEC1. A.J.W. and S.A. wrote the manuscript with input from all authors.

## Additional information

**Competing interests:** S.R.K. is an equity holder of TwinStrand BioSciences Inc., a for profit entity that is commercializing Duplex Sequencing. TwinStrand BioSciences Inc. had no role in designing or executing the work described in this manuscript. The remaining authors declare no competing interests.

