## [Peer Review File · Nature Communications]

Reviewers' comments:

Reviewer # 1 (Remarks to the Author):

This is a nicely-written paper that convincingly introduces a new mitochondrial mutator system in *Drosophila* and compares it to the classic POLG model in terms of lifespan, mtDNA mutations and mitochondrial function. The new model is likely to be of broad interest and utility in the field.

Minor point: the choice of substrates for the mitochondrial experiments is odd, and more appropriate for mammalian mitochondria. Mitochondria from flies use proline and glycerol phosphate much better than the glutamate+ malate and succinate used here, so these substrates would have been better able to show up functional deficits than the ones chosen.

Reviewer # 2 (Remarks to the Author):

This is an extremely interesting paper which potentially opens up the field of mitochondrial mutagenesis. The three most important findings are that the originally created *tamas* (*Polg*) mutator, which in *Drosophila* is recessive lethal, has a minimal phenotypic effect as a heterozygote, whereas the implementation of the human *APOBEC1* gene, which encodes an RNA editing enzyme that also acts as a C to T mutator in DNA, produces a severe and typically 'mitochondrial' phenotype of respiratory chain impairment, disturbed locomotor function and decreased lifespan. The third finding concerns the parent-of-origin effect on the mutation load conferred by the *tamas* (but not the *mito-APOBEC1*) mutator, being much stronger in the female line.

The data in this paper appear to be sound, and proper controls have been implemented. However, I feel that the authors' interpretations are a bit too narrow and should be backed up with additional data and modified if needed. After these issues are fixed (see below), I have no doubt that this will be a significant contribution to the field that deserves to be published in NC.

Major issues

1. The major phenotypic difference between the *tamas* mutator and the *mito-APOBEC1* expressing flies to me suggests the possibility that the latter has additional effects on mitochondrial functions distinct from those caused by the mutation burden per se. One obvious thing that should be tested is effects on mitochondrial RNA levels, since the presence of a considerable number of abasic sites and/or U residues over the genome could limit the overall rate of transcription (mismatches should be tolerated by the transcription machinery but abasic sites in the template strand or even just the presence of the BER machinery might not). The authors should therefore measure the levels of at least a few representative transcripts from each strand of the major gene clusters using quantitative RT-PCR, versus controls.

2. Although the authors have checked effects on mtDNA copy number by PCR, this does not distinguish bona fide mtDNA from incompletely or abortively replicated molecules. It would thus be helpful to include a Southern blot analysis alongside to test whether mtDNA is still intact in the *mito-APOBEC1* strain. There may also be effects on mtDNA replication which could partially underlie the phenotype. Stalled, regressed or degraded forks should be easily detected on 2D agarose gels (the entire genome need not be interrogated, just one or two restriction fragments covering e.g. the replication origin and a region distant from it).

3. In the light of these doubts, and even if the additional experiments fully support the authors' original interpretations, I think it is not helpful to the field to use a stridently worded interpretation as the title of the paper. Instead, something like '*APOBEC1* expression in *Drosophila* mitochondria acts as a potent mtDNA mutator and produces a phenotype of mitochondrial dysfunction' would be a title that more accurately fits the main findings and does not make definitive claims about the mechanism of this effect, which remain to be fully verified.

Minor issues

4. An important corollary of the findings is that they cast some doubt on the assumption that the developmental lethality of the *tamas* mutator when homozygous is really due to mutagenesis, and not to some other effect of the *tamas* allele on mtDNA replication. I feel there is no need for the authors to delve into this point mechanistically, but it should nevertheless appear in the Discussion.

5. The authors already point out that the type of mutation produced by APOBEC1 is likely to be damaging because of the high A+T content of the *Drosophila* mitochondrial genome, with C/G pairs confined to critical sites. This point could be better fleshed out by explaining that almost all third-base positions in the coding sequence are either A or T, whilst C and G are found predominantly in first or second base positions, giving rise to a high frequency of nonsynonymous changes brought about by mito-APOBEC1. Can the expected patterns of substitutions be modelled computationally, to test how far the specific mutational patterns of the two mutators fit these predictions? One of the 'unknowns' may be the frequency of different types of error introduced by Pol gamma, which such an analysis might reveal.

6. The parent-of-origin difference in the mutation levels conferred by the heterozygous *tamas* mutator is consistent with the literature on the developmental effects of severe or even null alleles of *tamas*, which give lethality only at L3 stage. This supports the idea that the store of maternally supplied Pol gamma is sufficient to carry the zygote far into development. I suggest to add this point and cite the relevant literature.

7. Although the A+T region cannot be reliably sequenced by NGS methods, it can be sequenced using conventional Sanger sequencing, though I do not see any specific advantage in carrying out such a laborious analysis at this point. However, the relevant figure legends should note that the non-coding A+T region is excluded from the analysis, or at least specify it, as 'mtDNA coding-region mutation load' etc. Even though this is mentioned in the Materials and Methods, a casual reader might not notice.

8. The post-hoc correction to ANOVA is by Tukey, not Turkey.

9. The APOBEC1 signal in the Western of Fig. 2B is obviously poor. Whilst I see nothing 'suspicious' about this, it would be good to substitute a better image for publication, especially given the much cleaner signals in Fig. 2C.

10. Although it is already stated in M&M, it's appropriate to mention in the legend to Fig. 3 that the assays were conducted on males.

Reviewer # 3 (Remarks to the Author):

In their manuscript, "A new mitochondrial DNA mutator model shows that quality not quantity of mutations affects organismal fitness in *Drosophila*", Whitworth and colleagues investigate the role of mutation of the mitochondrial genome in aging of the fly. While the accumulation of mutations in mitochondrial DNA has been linked to aging in mammalian systems, Andreazza et al. show that an exonuclease deficient DNA polymerase gamma, which replicates the mitochondrial genome, induces mutation, however, unlike mouse model systems, this increase in mutagenesis is not associated with an aging phenotype or mitochondrial dysfunction. This result is consistent with a previous recent publication showing that mutants of the *Drosophila* DNA pol gamma fail to induce an aging phenotype (Kauppila et al., PNAS, 2018). Andreazza et al., subsequently produce a novel mitochondrial mutator model by expressing the cytidine deaminase APOBEC1 specifically in the mitochondrial. APOBEC1 expression produces similar levels of mutation as the pol gamma exonuclease mutant, however, APOBEC1 expression in the mitochondria induces mitochondrial dysfunction and an aging phenotype. The authors analyze the mutation spectrum induced by APOBEC1 compared to the pol gamma mutant and determine that APOBEC1 expression induces exclusively C to T substitutions. They also find that APOBEC1-induced mutations are more likely to produce non-synonymous changes, which the authors argue is the likely cause of the

mitochondrial dysfunction and aging.

Overall, the work is well controlled and interesting. The lack of an aging phenotype in flies expressing active APOBEC1 that is not targeted to the mitochondria or catalytically inactive mito-APOBEC1 clearly indicate that the aging phenotype in these flies is due to active deamination of the mitochondrial genome. However, I believe addressing the following comments will strengthen the manuscript.

- 1) In Fig. 1, there appears to be a 4 fold higher load of C:G transitions (y-axis scale is 10^{-3}) in panel D compared to the point mutation load (y-axis scale is 10^{-4}) in panel A. This needs to be explained because the transitions should be a subset of the point mutation so the data in panel D should be lower than in panel A.
- 2) Please include the units for mutation load in the fig. 1. Mutation load could mean total mutations or mutation density (mutations per nucleotide), which is used here.
- 3) In lines 275-277, the authors state that the high ratio of NS to S mutations is due to mito-APOBEC1 causing exclusively C:G transitions and then suggest this underlies the aging phenotype. This effect would presumably be due to inactivation of mitochondrial proteins through the NS substitutions. However, in Fig 1 the majority of point mutations in the exo- pol gamma flies are also C:G transitions. There is less than a 2-fold difference in C:G transition load between the exo- pol gamma flies and the mito-APOBEC1 flies. That difference does not seem enough to account for the difference in the NS:S substitution ratio.
- 4) In relation to point 3, APOBEC1 deaminates DNA in a sequence specific manner, targeting TC dinucleotides. Could this sequence specificity be contributing to the elevated NS:S ratio? A comparison of the sequence context of the mutations in the exo- pol gamma and mito-APOBEC1 would address this.
- 5) Mito-APOBEC1 mutations also appear to be strand biased. Based on an analysis of the mutations in supplemental table 1, 3-4 fold more G mutations occur than C mutations in the mito-APOBEC1 flies. Exo- pol gamma flies have more C mutations than G mutations. This indicates that APOBEC1 is deaminating one of the DNA strands of the mitochondrial DNA more than the other strand. Since APOBEC1 is ssDNA specific, is there any idea what the ssDNA target of the enzyme is in this system? In lines 327-328, the authors speculate mito-APOBEC1 deaminates transcription intermediates. The analysis of strand bias, could answer this question. Could it also contribute to the NS:S ratio?
- 6) In the discussion, I believe it would be useful for the authors to discuss why no aging phenotype is observed in the exo- pol gamma flies when it induces a significant number of insertion/deletion mutations that would be as damaging (if not more) as the C:G transition mutations.
- 7) While the authors nicely show no difference occurs in the number of molecules of mitochondrial DNA upon expression of mito-APOBEC1 or exo- pol gamma, could the two mutators differentially impact the removal of defective mitochondria by mitophagy? Selective removal of mitochondrial genomes containing deletions has been suggested in at least one other publication (Kandul, et al., Nature Communications, 2016).
- 8) Presumably other mutators with less specificity than the mito-APOBEC1, could also induce an aging phenotype if they produced enough mutations to accumulate the same levels of C:G transitions. Would that level of mutagenesis be physiologically relevant to make the mito-APOBEC1 an effective model for more relevant mutagens that would be expected to act in a non-engineered mitochondria?

Response to reviewers' comments:

We thank the reviewers for their positive support for this study and their constructive critiques. In line with their suggestions we have amended the text to clarify our methodology, meaning or intention, and we have added several new analyses which together we feel further strengthen the study.

Reviewer #1 (Remarks to the Author):

This is a nicely-written paper that convincingly introduces a new mitochondrial mutator system in *Drosophila* and compares it to the classic POLG model in terms of lifespan, mtDNA mutations and mitochondrial function. The new model is likely to be of broad interest and utility in the field.

Minor point: the choice of substrates for the mitochondrial experiments is odd, and more appropriate for mammalian mitochondria. Mitochondria from flies use proline and glycerol phosphate much better than the glutamate+malate and succinate used here, so these substrates would have been better able to show up functional deficits than the ones chosen.

We agree that the precise details of the protocol are more typical for mammalian than insect mitochondrial physiology but while fly mitochondria do indeed utilise proline more efficiently than glutamate+malate (likely due to free permeability of fly mitochondria to proline, although this seems to be unresolved), we consider that any potential difference is probably offset somewhat in our assay by conducting the analysis at 30°C. Nevertheless, we appreciate the reviewer's point and will adapt our protocols in future to maximise the potential to detect small differences. In the context of the current study, we are confident that we have detected robust differences where they are present and the lack of observable phenotypes is a true reflection of the condition.

Reviewer #2 (Remarks to the Author):

This is an extremely interesting paper which potentially opens up the field of mitochondrial mutagenesis. The three most important findings are that the originally created *tamas* (*Polg*) mutator, which in *Drosophila* is recessive lethal, has a minimal phenotypic effect as a heterozygote, whereas the implementation of the human *APOBEC1* gene, which encodes an RNA editing enzyme that also acts as a C to T mutator in DNA, produces a severe and typically 'mitochondrial' phenotype of respiratory chain impairment, disturbed locomotor function and decreased lifespan. The third finding concerns the parent-of-origin effect on the mutation load conferred by the *tamas* (but not the mito-*APOBEC1*) mutator, being much stronger in the female line.

The data in this paper appear to be sound, and proper controls have been implemented. However, I feel that the authors' interpretations are a bit too narrow and should be backed up with additional data and modified if needed. After these issues are fixed (see below), I have no doubt that this will be a significant contribution to the field that deserves to be published in NC.

Major issues

1. The major phenotypic difference between the *tamas* mutator and the mito-APOBEC1 expressing flies to me suggests the possibility that the latter has additional effects on mitochondrial functions distinct from those caused by the mutation burden per se. One obvious thing that should be tested is effects on mitochondrial RNA levels, since the presence of a considerable number of abasic sites and/or U residues over the genome could limit the overall rate of transcription (mismatches should be tolerated by the transcription machinery but abasic sites in the template strand or even just the presence of the BER machinery might not). The authors should therefore measure the levels of at least a few representative transcripts from each strand of the major gene clusters using quantitative RT-PCR, versus controls.

This is a good suggestion. We have now analysed by qRT-PCR the expression levels of several mitochondrial transcripts across the genome. Specifically, we have analysed 4 transcripts; *Cyt-b*, *ND3* and *CoI* from the major strand and *ND4* from the minor strand. Interestingly, these results show a consistent decrease in transcript levels for the non-mutating mito-APOBEC1[E63A] control while only *Cyt-b* showed a significant change compared to control in mito-APOBEC1 flies, which was in fact increased. These data are shown in new Fig. 3f. We interpret these results to indicate that mito-APOBEC1 is not causing a generalised or gross disruption of mitochondrial transcription, and any small changes observed are unlikely to significantly contribute to the organismal phenotypes.

2. Although the authors have checked effects on mtDNA copy number by PCR, this does not distinguish bona fide mtDNA from incompletely or abortively replicated molecules. It would thus be helpful to include a Southern blot analysis alongside to test whether mtDNA is still intact in the mito-APOBEC1 strain. There may also be effects on mtDNA replication which could partially underlie the phenotype. Stalled, regressed or degraded forks should be easily detected on 2D agarose gels (the entire genome need not be interrogated, just one or two restriction fragments covering e.g. the replication origin and a region distant from it).

We appreciate the value of this analysis as we are keen to determine as much as possible whether the phenotypes we see are principally due to point mutations. As suggested, we have analysed the mitochondrial genome integrity in the mito-APOBEC1 model (and controls) by Southern blot analysis, and further complemented this by long-range PCR. Encouragingly, neither of these approaches revealed any evidence of mtDNA with deletions. However, quantification of the Southern blots revealed a small but significant decrease in mtDNA levels. These data are shown in new Fig. 3c-e. These results led us to consider our previous qPCR analysis of mtDNA levels which had shown a consistent trend towards lower levels, although it appeared that the technical variability rendered these differences non-significant. Reassessing these data, we realised that we had used a more conservative statistical analysis than was necessary (no matched comparisons).

Applying a more appropriate and lenient analysis (matched comparisons), the mtDNA levels in mito-APOBEC1 and mito-APOBEC1[E63A] 20-day-old flies are significantly decreased, in line with the Southern blot data. These results have been amended in the respective figures and text. Although some of these differences are statistically meaningful, changes are consistently modest in size. Thus, considering the weight of evidence from the range of techniques used (qPCR, Southern blot, long-range PCR), in conjunction with the sequence and transcript analysis, we believe that little if any contribution to the organismal phenotypes comes from disruption of other aspects of mitochondrial genome than the mtDNA mutations directly.

3. In the light of these doubts, and even if the additional experiments fully support the authors' original interpretations, I think it is not helpful to the field to use a stridently worded interpretation as the title of the paper. Instead, something like 'APOBEC1 expression in *Drosophila* mitochondria acts as a potent mtDNA mutator and produces a phenotype of mitochondrial dysfunction' would be a title that more accurately fits the main findings and does not make definitive claims about the mechanism of this effect, which remain to be fully verified.

We understand the reviewer's point of view but there were a number of key elements that we wanted to convey in our title. One was that this was a novel type of mtDNA mutator model, since, as discussed, the field has been heavily reliant on the classic POLG mutator. Second, we wanted to make a clear counter-point to the recent PNAS paper from Kauppila et al. entitled "Mutations of mitochondrial DNA are not major contributors to aging of fruit flies" which is in stark contrast to our study. It is clear from our study that mtDNA mutations are in fact capable of influencing *Drosophila* lifespan, but it depends on the nature of the mutations (and the model). While the reviewer's proposed title is certainly adequate, this outcome would not be reflected in that title.

Moreover, we also wanted to convey what we consider is a very important outcome of our analysis; that is, a much more profound and accurate insight into the nature of mtDNA mutations is gained from a much more detailed analysis of the exact nature of the observed mutations. It is only through this deep analysis of the mutations that we have been able to rationalise why two models with an apparently similar mutational burden can have such dramatically different organismal consequences. This clearly indicates that it isn't the quantity of mtDNA mutations that specifically affects organismal fitness but what those mutations are (i.e. their quality). We feel that this take-home message needs the prominent visibility afforded in the title.

Minor issues

4. An important corollary of the findings is that they cast some doubt on the assumption that the developmental lethality of the *tamas* mutator when homozygous is really due to mutagenesis, and not to some other effect of the *tamasexo*- allele on mtDNA replication. I feel there is no need for the authors to delve into this point mechanistically, but it should nevertheless appear in the Discussion.

We have added a comment in the Discussion on this, and cited the recent paper of Samstag et al. (PLoS Genetics, 2018) who have also discussed this matter.

5. The authors already point out that the type of mutation produced by APOBEC1 is likely to be damaging because of the high A+T content of the *Drosophila* mitochondrial genome, with C/G pairs confined to critical sites. This point could be better fleshed out by explaining that almost all third-base positions in the coding sequence are either A or T, whilst C and G are found predominantly in first or second base positions, giving rise to a high frequency of nonsynonymous changes brought about by mito-APOBEC1. Can the expected patterns of substitutions be modelled computationally, to test how far the specific mutational patterns of the two mutators fit these predictions? One of the 'unknowns' may be the frequency of different types of error introduced by Pol gamma, which such an analysis might reveal.

In our original manuscript we calculated the C:G content in first, second, and third codon positions for each protein within the mitochondrial genome, and demonstrated the depletion of C:G sites in the third codon position (new Supplementary Fig. 6c). We have now expanded upon this analysis in three ways:

- 1) We characterised the prevalence of mutations in first, second, and third codon positions in mito-APOBEC1 and maternal *tam*^{exo-} mutator models, demonstrating that mito-APOBEC1 flies show an enrichment in first and second codon position mutations relative to *tam*^{exo-} flies (Supplementary Fig. 6d).
- 2) We analysed the theoretical rate of non-synonymous vs synonymous substitutions for each mutation type throughout the protein-coding sequence to clarify that the predominant mutation type observed in mito-APOBEC1 flies frequently results in non-synonymous substitutions (Supplementary Fig. 6e).
- 3) We have analysed the mutation load within each trinucleotide context (Supplementary Fig. 7). Together, these data support the model that mito-APOBEC1 flies bear an enrichment in first and second codon position mutations in a highly specific genomic context. Conversely, *tam*^{exo-} flies display a diverse mutation spectrum, including a much greater proportion of mutations in third codon positions and a large number frequently benign T:A>C:G mutations. These mutations are more likely to be synonymous substitutions, influencing the skewed NS:S ratio we previously discussed. We have added these new analyses to the manuscript and discussed their implications in context.

6. The parent-of-origin difference in the mutation levels conferred by the heterozygous *tamas* mutator is consistent with the literature on the developmental effects of severe or even null alleles of *tamas*, which give lethality only at L3 stage. This supports the idea that the store of maternally supplied Pol gamma is sufficient to carry the zygote far into development. I suggest to add this point and cite the relevant literature.

We have added a comment and citation in the Discussion to this effect.

7. Although the A+T region cannot be reliably sequenced by NGS methods, it can be sequenced using conventional Sanger sequencing, though I do not see any specific advantage in carrying out such a laborious analysis at this point. However, the relevant figure legends should note that the non-coding A+T region is excluded from the analysis, or at least specify it, as 'mtDNA coding-region mutation load' etc. Even though this is mentioned in the Materials and Methods, a casual reader might not notice.

We have added this clarification in the relevant Methods and reiterated it in the first figure legend.

8. The post-hoc correction to ANOVA is by Tukey, not Turkey.

Corrected.

9. The APOBEC1 signal in the Western of Fig. 2B is obviously poor. Whilst I see nothing 'suspicious' about this, it would be good to substitute a better image for publication, especially given the much cleaner signals in Fig. 2C.

We understand the reviewer's point but this level of signal is an accurate reflection of the relative level of expression. The purpose of showing this blot is to verify that a single protein species is produced as expected, and to give a sense of the relative expression levels both between the two mito-APOBEC1 transgenes and to the control mito-HA-GFP transgene. The conditions used for this blot also affords comparisons between mito- and cyto-APOBEC1 transgenes, as the same control, da>mito-HA-GFP, is used. Overexposure of blot in Fig. 2b would make comparison with cyto-APOBEC1 (Supplementary Fig. 3b) unreasonable. This is an important comparison as it demonstrates that even though the cyto-APOBEC1 expression is benign, protein levels are much higher than mito-APOBEC1. In summary, we feel that this blot provides a true reflection of the relatively low levels of expression from these transgenes, and offers appropriate comparisons where necessary.

10. Although it is already stated in M&M, it's appropriate to mention in the legend to Fig. 3 that the assays were conducted on males.

This has been added.

Reviewer #3 (Remarks to the Author):

In their manuscript, "A new mitochondrial DNA mutator model shows that quality not quantity of mutations affects organismal fitness in *Drosophila*, Whitworth and colleagues investigate the role of mutation of the mitochondrial genome in aging of the fly. While the accumulation of mutations in mitochondrial DNA has been linked to aging in mammalian systems, Andreatza et al. show that an exonuclease deficient DNA polymerase gamma, which replicates the mitochondrial genome,

induces mutation, however, unlike mouse model systems, this increase in mutagenesis is not associated with an aging phenotype or mitochondrial dysfunction. This result is consistent with a previous recent publication showing that mutants of the *Drosophila* DNA pol gamma fail to induce an aging phenotype (Kauppila et al., PNAS, 2018). Andreazza et al., subsequently produce a novel mitochondrial mutator model by expressing the cytidine deaminase APOBEC1 specifically in the mitochondrial. APOBEC1 expression produces similar levels of mutation as the pol gamma exonuclease mutant, however, APOBEC1 expression in the mitochondria induces mitochondrial dysfunction and an aging phenotype. The authors analyze the mutation spectrum induced by APOBEC1 compared to the pol gamma mutant and determine that APOBEC1 expression induces exclusively C to T substitutions. They also find that APOBEC1-induced mutations are more likely to produce non-synonymous changes, which the authors argue is the likely cause of the mitochondrial dysfunction and aging.

Overall, the work is well controlled and interesting. The lack of an aging phenotype in flies expressing active APOBEC1 that is not targeted to the mitochondria or catalytically inactive mito-APOBEC1 clearly indicate that the aging phenotype in these flies is due to active deamination of the mitochondrial genome. However, I believe addressing the following comments will strengthen the manuscript.

1) In Fig. 1, there appears to be a 4 fold higher load of C:G transitions (y-axis scale is 10^{-3}) in panel D compared to the point mutation load (y-axis scale is 10^{-4}) in panel A. This needs to be explained because the transitions should be a subset of the point mutation so the data in panel D should be lower than in panel A.

We calculate mutation frequencies across the whole genome as the total number of mutations identified, including multiple counts for higher heteroplasmies, divided by the total coverage of bases sequenced. For calculations that concern specific genomic subsets, e.g. by base type (Fig. 1d and Supplementary Fig. 5a) or genomic feature (Supplementary Fig. 5b and Supplementary Fig. 7), the denominator is determined by the subset category indicated on the X-axis (e.g. number of sequenced Cs, Gs, etc; or sequenced protein coding bases; or number of sequenced bases in a given codon). Specifically regarding the mutation load in Fig. 1d, as C:G transitions are only possible at C:G sites in the genome, only C:G sites are tabulated for the denominator of this calculation. So, the frequencies shown in Fig. 1a are not intended to be, and indeed should not be, directly compared to those in Fig. 1d. We have further clarified this in the main text.

2) Please include the units for mutation load in the fig. 1. Mutation load could mean total mutations or mutation density (mutations per nucleotide), which is used here.

We agree with the reviewer that the term “mutation load” is ambiguous. We have changed this term to “mutation frequency” throughout the revised manuscript and figures, and the added text describing the calculation of mutation frequency should help to clarify this for the readership.

3) In lines 275-277, the authors state that the high ratio of NS to S mutations is due to mito-APOBEC1 causing exclusively C:G transitions and then suggest this underlies the aging phenotype. This effect would presumably be due to inactivation of mitochondrial proteins through the NS substitutions. However, in Fig 1 the majority of point mutations in the *exo-pol gamma* flies are also C:G transitions. There is less than a 2-fold difference in C:G transition load between the *exo-pol gamma* flies and the mito-APOBEC1 flies. That difference does not seem enough to account for the difference in the NS:S substitution ratio.

We have clarified and expanded upon this hypothesis with a more thorough characterization of the mutational spectra of these flies. As explained in our response to Reviewer 2 above, we have now shown that there is an almost total lack of third codon mutations in mito-APOBEC1 flies (new Supplementary Fig. 6d), whereas *tam^{exo-}* still has a substantial proportion of mutations in the third codon position. This likely has a considerable impact on the occurrence of NS mutations and the reason that the NS:S ratio shown in Fig. 7a, b is much lower for *tam^{exo-}* than for mito-APOBEC1.

4) In relation to point 3, APOBEC1 deaminates DNA in a sequence specific manner, targeting TC dinucleotides. Could this sequence specificity be contributing to the elevated NS:S ratio? A comparison of the sequence context of the mutations in the *exo-pol gamma* and mito-APOBEC1 would address this.

We have further characterised our mutator lines by analysing the mutations with respect to the trinucleotide (rather than dinucleotide) spectrum. The distribution of the number of mutations across all possible trinucleotide sequences reveals that while *tam^{exo-}* mutates in a wide variety of genomic contexts, mito-APOBEC1 mutates in a highly specific dinucleotide context (TCn), particularly at TCT sites. It is possible that this specificity contributes to the difference in NS:S ratios observed between the two strains but it has not been explored further. These data are presented in new Supplementary Fig. 7.

5) Mito-APOBEC1 mutations also appear to be strand biased. Based on an analysis of the mutations in supplemental table 1, 3-4 fold more G mutations occur than C mutations in the mito-APOBEC1 flies. *Exo-pol gamma* flies have more C mutations than G mutations. This indicates that APOBEC1 is deaminating one of the DNA strands of the mitochondrial DNA more than the other strand. Since APOBEC1 is ssDNA specific, is there any idea what the ssDNA target of the enzyme is in this system? In lines 327-328, the authors speculate mito-APOBEC1 deaminates transcription intermediates. The analysis of strand bias, could answer this question. Could it also contribute to the NS:S ratio?

This is a very interesting phenomenon. We have now more deeply analysed the strand bias of the mito-APOBEC1 mutator and found that there is indeed a dramatic bias (new Supplementary Fig. 2a). In fact, there is a nearly 20-fold difference in mutation rate between the major and minor

strands; the minor strand being much more mutated. The reason for this striking strand bias is unknown and mapping the mutations provided no obvious clues (new Supplementary Fig. 2b). We speculate the minor strand, whose replication is discontinuous (lagging strand), may be more vulnerable to accumulating mutations. Moreover, current literature also supports some role for APOBEC1 in mutating during transcription (Lada et al., PLoS Genetics, 2015); this has been cited in the discussion. It will be interesting to resolve the mechanistic cause in subsequent studies.

6) In the discussion, I believe it would be useful for the authors to discuss why no aging phenotype is observed in the *exo-pol gamma* flies when it induces a significant number of insertion/deletion mutations that would be as damaging (if not more) as the C:G transition mutations.

Our sequencing analysis shows that when indels are present, they rarely exist in heteroplasmies above 10% (Supplementary Table 2). Thus, the relatively limited occurrence of indel mutations is likely readily complemented by wild-type molecules.

7) While the authors nicely show no difference occurs in the number of molecules of mitochondrial DNA upon expression of *mito-APOBEC1* or *exo-pol gamma*, could the two mutators differentially impact the removal of defective mitochondria by mitophagy? Selective removal of mitochondrial genomes containing deletions has been suggested in at least one other publication (Kandul, et al., Nature Communications, 2016).

This is, at least theoretically, possible but we have no data at this stage to suggest that the two mutator models are inducing mitochondrial quality control processes differently. This is the subject of ongoing work and beyond the scope of the current study.

8) Presumably other mutators with less specificity than the *mito-APOBEC1*, could also induce an aging phenotype if they produced enough mutations to accumulate the same levels of C:G transitions. Would that level of mutagenesis be physiologically relevant to make the *mito-APOBEC1* an effective model for more relevant mutagens that would be expected to act in a non-engineered mitochondria?

We would agree that other mutagens could indeed induce an ageing phenotype if enough 'damaging' (i.e. NS) mutations arose. However, a less specific mutagen that the reviewer conceives of would also presumably need to contend with a relatively higher amount of mutations in non-coding genomic elements including tRNA and rRNA which may lead to a more generalised and possibly more detrimental mitochondrial disruption.

REVIEWERS' COMMENTS:

Reviewer # 1 (Remarks to the Author):

It is not adequate to just tell the reviewer in a cover letter that using the wrong substrates for respiration assays is OK. The authors need to include this recognition of use of inappropriate substrates explicitly in the methods section, else others may be tempted to use the same substrates thinking it's OK.

Reviewer # 2 (Remarks to the Author):

My criticisms have been satisfactorily addressed, with the exception of the point about the article title. I do agree that the title of the Kauppila article is misleading, but anyhow, the authors have argued the point and I would obviously not wish to block publication on what is ultimately an editorial issue. So I am happy to leave this to the editor and authors to agree.

Reviewer # 3 (Remarks to the Author):

The authors have adequately addressed my prior comments. I support publication of the work in Nature Communications.

My one remaining suggestion is for the authors to consider replacing "mutation frequency" with the term "mutation density" to differentiate their method of standardizing mutations by the available target sequences from other mutation based experiments that utilize mutation frequency to mean the frequency of a mutant among a surviving cell population.

Response to Reviewer comments

Reviewer #1 (Remarks to the Author):

It is not adequate to just tell the reviewer in a cover letter that using the wrong substrates for respiration assays is OK. The authors need to include this recognition of use of inappropriate substrates explicitly in the methods section, else others may be tempted to use the same substrates thinking it's OK.

It is inaccurate and misleading to describe the substrates as “inappropriate” or “wrong”. As we agree that upon reflection the conditions used were not ideal but, nevertheless, the specific assay conditions do indeed report on the nature of mitochondrial respiration intended, albeit they may be sub-optimal. However, to indicate that more optimal protocols are available we have included a statement to this effect in the method section.

Reviewer #2 (Remarks to the Author):

My criticisms have been satisfactorily addressed, with the exception of the point about the article title. I do agree that the title of the Kauppila article is misleading, but anyhow, the authors have argued the point and I would obviously not wish to block publication on what is ultimately an editorial issue. So I am happy to leave this to the editor and authors to agree.

We have amended the title to be less strident.

Reviewer #3 (Remarks to the Author):

The authors have adequately addressed my prior comments. I support publication of the work in Nature Communications.

My one remaining suggestion is for the authors to consider replacing "mutation frequency" with the term "mutation density" to differentiate their method of standardizing mutations by the available target sequences from other mutation based experiments that utilize mutation frequency to mean the frequency of a mutant among a surviving cell population.

We appreciate that there is lack of consistency in the nomenclature describing the occurrence and abundance of mtDNA mutations, largely due to a lack of unifying methodology interrogating such mutations. Consequently, we have elaborated precisely how we determine mutations and how we derive the parameters with which we report them. Since mutation density is not a commonly used term in the field, and may add further confusion, we have opted to use “mutation load” as this term has been recently used in the context of both unique and total mtDNA mutations.